# Functional divergence of CYP76AKs shapes the chemodiversity of abietane-type diterpenoids in genus *Salvia*

Jiadong Hu[1,2,7], Shi Qiu ®[1,7] ✉, Feiyan Wang[1,7], Qing Li[2,7], Chun-Lei Xiang[3], Peng Di[4], Ziding Wu[1], Rui Jiang[1], Jinxing Li[1], Zhen Zeng[1], Jing Wang[1], Xingxing Wang[1], Yuchen Zhang[1], Shiyuan Fang[1], Yuqi Qiao[1], Jie Ding[5], Yun Jiang[5], Zhichao Xu ®[6] ✉, Junfeng Chen ®[1] ✉ & Wansheng Chen ®[1,2] ✉

The genus *Salvia* L. (Lamiaceae) comprises myriad distinct medicinal herbs, with terpenoids as one of their major active chemical groups. Abietane-type diterpenoids (ATDs), such as tanshinones and carnosic acids, are specific to *Salvia* and exhibit taxonomic chemical diversity among lineages. To elucidate how ATD chemical diversity evolved, we carried out large-scale metabolic and phylogenetic analyses of 71 *Salvia* species, combined with enzyme function, ancestral sequence and chemical trait reconstruction, and comparative genomics experiments. This integrated approach showed that the lineage-wide ATD diversities in *Salvia* were induced by differences in the oxidation of the terpenoid skeleton at C-20, which was caused by the functional divergence of the cytochrome P450 subfamily CYP76AK. These findings present a unique pattern of chemical diversity in plants that was shaped by the loss of enzyme activity and associated catalytic pathways.

Plant-derived natural products are a valuable resource for pharmacological research and the development of health-related products, and their structural diversity is related to various biological activities[1]. Both Taxol, a diterpenoid alkaloid obtained from *Taxus chinensis* (Pilg.) Rehder, and Artemisinin, a sesquiterpenoid derived from *Artemisia annua* L., have received a great deal of attention worldwide due to their exceptional anticancer and antimalarial effects, respectively[2,3]. Because of the high demands for natural products, approaches for obtaining specific compounds via metabolic engineering have become a major trend, necessitating the elucidation of biosynthetic pathways and the identification of key genes involved in these pathways.

Most natural products obtained from plants are specialized metabolites. However, elucidating a biosynthetic pathway is limited when the pathway is complex and includes multiple steps, making it challenging to identify the relevant genes[4]. Generally, the emergence of plant specialized metabolites is an evolved defense mechanism that protects plants against a barrage of biotic and abiotic stresses, with chemical diversity reflected in the diversity of their skeletons and chemical modifications[5]. In general, similar skeletons are derived from similar biosynthetic pathways within homologous plants. However, natural products based on the same skeletons with various chemical modifications (e.g., glycosylation, methylation, hydroxylation, acylation, prenylation) are diverse within species. Such structural modifications play important roles in how plants respond to changes in the external environment for their growth and development[6].

[1]The SATCM Key Laboratory for New Resources & Quality Evaluation of Chinese Medicine, Institute of Chinese Materia Medica, Shanghai University of Traditional Chinese Medicine, Shanghai 201203, China. [2]Department of Pharmacy, Second Affiliated Hospital of Navy Medical University, Shanghai 200003, China. [3]CAS Key Laboratory for Plant Diversity and Biogeography of East Asia, Kunming Institute of Botany, Chinese Academy of Sciences, Kunming 650201, China. [4]State Local Joint Engineering Research Center of Ginseng Breeding and Application, College of Chinese Medicinal Materials, Jilin Agricultural University, Changchun 130118, China. [5]Urban Horticulture Research and Extension Center, Shanghai Chenshan Botanical Garden, Shanghai 201602, China. [6]College of Life Sciences, Northeast Forestry University, Harbin 150040, China. [7]These authors contributed equally: Jiadong Hu, Shi Qiu, Feiyan Wang, Qing Li. ✉e-mail: davidhugh@msn.cn; zcxu@nefu.edu.cn; 0000002928@shutcm.edu.cn; chenwansheng@smmu.edu.cn

Given the rapid advancements in recent omics technologies, it is now possible to trace evolutionary origin, distribution, and composition of metabolites on lineage scale[7], and shed light on their ancient or novel functions, as well as the emergence and radiation of their biosynthetic pathways over time[8] For instance, the Mint Plant Genome Project has made significant contributions in this regard, by investigating the distribution patterns of volatile terpenoids in the Lamiaceae family[9], and exploring the radiations of the iridoid pathway across various lineages[10]. They have also brought insights into the mechanisms behind the loss and subsequent re-evolution of nepetalactone biosynthesis in the *Nepeta* lineage[11]. Another highly representative instance is the origin of morphinan in genus *Papaver* (Papaveraceae). By tracing the fusion events of *STORR* gene among *Papaver* genomes, and correlated whole-genome duplications (WGDs), chromosomal rearrangements, and subgenome evolution[12–14], the innovation of this specialized metabolite was demonstrated. Therefore, establishing the genetic mechanism for the formation of natural product diversity across species is of great importance for understanding the pathways, providing synthetic elements for metabolic engineering and molecular breeding.

Tanshinones and carnosic acid–related compounds are specialized abietane-type diterpenoids (ATDs) in Lamiaceae[15,16]. Tanshinones were originally identified from *Salvia miltiorrhiza* Bunge and have been utilized for the treatment of coronary artery disease[17], angina pectoris[18], and myocardial infarction[19]. Carnosic acid and its derivatives found in *Salvia officinalis* L. have strong antioxidant[16], antiparasitic[20], antitumor[21], antifungal[22], antiadipogenic[23], and antibacterial[24] activities. The specific biosynthesis of these ATDs initiates from the biosynthesis of their common precursor, miltiradiene, catalyzing by copalyl diphosphate synthase (CPS) and kaurene synthase-like (KSL)[25]. In *S. miltiorrhiza*, CYP76AH1 catalyzes oxidation of miltiradiene at C-12 to form ferruginol[26]. Besides, other members of CYP76AH subfamily can catalyze oxidation of miltiradiene at C-7, −11, and −12 to generate a metabolite array with diverse modification on ATD rings[27–30]. CYP76AK subfamily further promotes the specificity and diversity of ATDs in *Salvia*. CYP76AK6 from *S. officinalis, Salvia fruticosa* Mill. *and Salvia pomifera* L., CYP76AK7 and CYP76AK8 from *Salvia rosmarinus* Schleid convert 11-hydroxyferruginol into carnosic acid by generating a carboxyl group at C-20[28–30], whereas CYP76AK1 from *S. miltiorrhiza* only executes hydroxylation at C-20 to generate an alcohol group[27]. This suggests that functional diversity of CYP76AKs may contribute to presence of different types of ATDs in various *Salvia* species. In addition, it is worth noting that the oxidation pattern present on the C-20 position could also play a crucial role in the formation of miltirone, the key precursor to tanshinones. This may involve the demethylation process (resulting in the loss of C-20) and the aromatization of ring B, which remain unresolved catalytic steps in the pathway (Supplementary Fig. 1). Therefore, a lineage-wide characterization of ATDs distribution, functions of CYP76AKs, as well as the relevant evolutionary events would be necessary for revealing the tanshinone biosynthesis pathway. Furthermore, in the case of evolution of CYP family in plants, the dynamic of CYP76AKs would provide insight into the interplay between function divergence and genome evolution in the evolution of CYP76AK clan[31,32].

*Salvia* L. (Lamiaceae, Nepetoideae)[33] is the largest genus in the mint family comprising ca. 1000 species[34], and it includes several culturally and economically important species that are used as traditional herbs (e.g., *S. miltiorrhiza* and *S. officinalis*)[35], fragrance (e.g., *Salvia sclarea* L.)[36], ornamentals (e.g., *Salvia splendens* Sellow ex Wied-Neuw.)[37], and functional foods (e.g., *Salvia hispanica* L.)[38]. *Salvia* has a cosmopolitan distribution consisted of three species diversity centers: Central-South America (ca. 500 species), Southwest Asia-Mediterranean (ca. 250 species), and East Asia (ca. 100 species)[34]. *Salvia* is known for having a rich variety of ATDs, making it a suitable model genus to investigate the evolutionary origins of ATDs

diversity[39]. Tanshinones primarily accumulate in *S. miltiorrhiza*[40], *Salvia przewalskii* Maxim.[41], and *Salvia yunnanensis* C.H.Wright[42], whereas carnosic acid–related metabolites have been found in European sage species such as *S. officinalis*[43], *S. fruticosa*[44], and *S. pomifera*[45]. It is, however, unclear whether the metabolic differences represent an isolated event or a consistent variation caused by clade differentiation within the genus. Thus, the overall correlated pattern between chemical diversity and genetic diversity based on a robust phylogeny of the genus is the basis support for investigating ATDs formation in this economically important genus.

Here, we combined the concepts of phylogenetic reconstruction, metabolome analysis, pathway analyses, and evolutionary mechanisms to characterize 71 *Salvia* species to investigate the molecular and evolutionary mechanisms that resulted in ATDs diversity in the genus. Our metabolome analysis revealed the distribution of ATDs in this genus, and in vitro enzyme activity assays were then performed to investigate the mechanisms by which the CYP76AK subfamily is involved in ATDs biosynthesis. In addition, an evolutionary model of the CYP76AK subfamily was proposed based on comparative genomics, phylogenetic analyses, and the enzymology of reconstructed ancestral biosynthetic enzymes. Our results suggest that the evolution of the CYP76AK family may have played a major role for the chemical diversity of ATDs in *Salvia*.

## Results

### Phylogenetic relationships within *Salvia*

To characterize chemical diversity and phylogenetic relationships within the genus *Salvia*, 77 species, including six outgroups (*Melissa officinalis* L., *Mentha spicata* L., *Clinopodium polycephalum* (Vaniot) C.Y.Wu & S.J.Hsuan, *Origanum vulgare* L., *Nepeta cataria* L., and *Prunella vulgaris* L.), were sampled for analyses (Supplementary Data 1), covering the main geographic distribution areas (America, West Asia, Europe, and East Asia) of the genus except Africa. In total, we generated 72 new transcriptomes, from mixed cDNA libraries of leaves and roots, resulting in an average of 40,876 transcripts representing 38.2 Mb per species (Supplementary Data 1). In addition to three published transcriptomes (*M. officinalis, M. spicata*, and *O. vulgare*)[9] and two genomes (*S. miltiorrhiza* and *N. cataria*)[11,46], 77 nuclear gene sets were obtained for our analyses. Five sets of orthologous groups (OGs), composing of 2178, 1532, 1169, 512, and 130 orthologous genes, respectively, were applied to reconstruct *Salvia* phylogeny using a coalescent method through a multi-step procedure with consideration for alignment length, species coverage, and other factors (Fig. 1; Supplementary Figs. 2–6). Monophyly of the genus *Salvia* and six subgenera (*Calosphace, Audibertia, Glutinaria, Sclarea*, "*Heterosphace*", and *Salvia*) were maximally supported (bootstrap support [BS] = 100%; Fig. 1; Supplementary Figs. 2-6). Consistent with previous phylogeny studies[47,48], the genus was split into three successive major clades (Clade I, II, and IV) that reflected their geographic distribution, which could be roughly divided into Eurasia, America, and East Asia. *Salvia abrotanoides* (Kar.) Sytsma and *S. rosmarinus*, belonging to subgenera *Perovskia* and *Rosmarinus* (PR), are sister to Clade I. Thus, the phylogeny of *Salvia* provided well-supported grouping for further lineage-wide metabolic profiling.

According to the tree constructed from 130 OGs concatenations for all taxa with four fossils as calibration points, the evolutionary timescale of *Salvia* lineages was further estimated (Fig. 1, Supplementary Fig. 7). Our chronogram illustrated the origin of the genus *Salvia* could be dated to the Late Eocene (~38.1 million years ago, Mya) to the Middle Oligocene (~26.1 Mya), for the crown group of *Salvia* were estimated to occurred at ~32.1 Mya with a 95% credible interval (CI) of 26.1–38.1. Shortly after that, the crown group of Clade I, *Rosmarinus*, and *Perovskia* might have occurred at ~29.5 Mya (95% CI: 23.7–35.2). In parallel, the node containing both Clade II and Clade IV might originate at ~28.2 Mya (95% CI: 22.7–33.8). Subsequently, Clade II

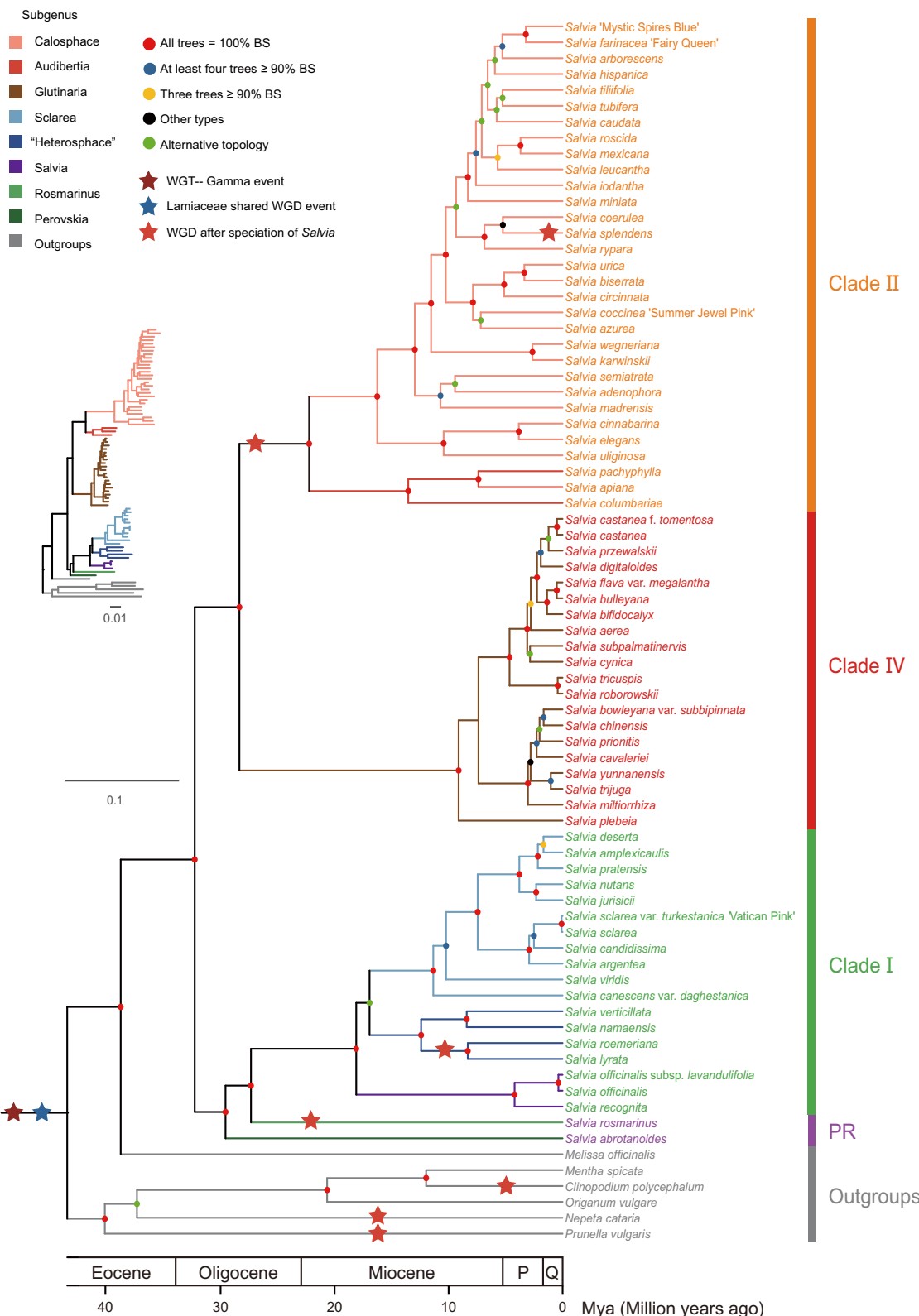

**Fig. 1 | Phylogenetic relationships of *Salvia* estimated by Astral using five gene sets.** Five levels of bootstrap values are denoted with red, blue, yellow, black, and green solid circles. Green solid circles indicate alternative topologies for the node among the five trees (see Supplementary Figs. 2–6). Species belonging to the same subgenus are marked with the same branch color. Species names are shown to the right of the tree, and clades are indicated on the far right and highlighted with colored labels. Colored solid stars denote genome duplication identified in this study. Mya million years ago, P Pliocene, Q Quaternary, WGD whole-genome duplication, WGT whole-genome triplication. PR subgenera *Perovskia* and *Rosmarinus*. Source data are provided as a Source Data file.

and Clade IV were estimated to diverge at ~22.1 Mya (95% CI: 17.6–27.0) and ~9.1 Mya (95% CI: 6.7–11.8), respectively.

Based on the transcriptomes and genome datasets in hand, we also identified whole-genome duplication (WGD) events among *Salvia* species according to synonymous substitutions per site ($K_S$) among paralogous gene pairs (Fig. 1; Supplementary Fig. 8). First, two previously known WGD events, the gamma event shared by core eudicots ($K_S = 2.27$ in average) and Lamiaceae shared WGD event ($K_S = 1.18$ in average)[37,49], were confirmed on our sampling scale. We didn't find a common WGD event among the whole *Salvia* genus, however, we identified a lineage-shared WGD ($K_S = 0.30$ on average) in Clade II. In addition, some individual species seemed to have undergone unique WGD events, i. e., *S. splendens*, *Salvia lyrata* L., and *Salvia roemeriana* Scheele.

## Chemical diversity and distribution of ATDs in *Salvia*

Understanding the relevancy between chemical diversity and genetic information is essential for the elucidation of biosynthetic pathways and the mining of functional genes. The roots and leaves of 71 *Salvia* species were prepared and analyzed using a liquid chromatography–mass spectrometry (LC-MS)–based metabolomics approach. Total ion current (TIC) chromatograms revealed that phenolic acids and ATDs were the main metabolites in both roots and leaves of *Salvia* (Supplementary Figs. 9–24). After peak detection, alignment and normalization using the MS-DIAL tools, the resultant feature tables were split into two tables based on the accumulation level of phenolic acids and ATDs and imported into SIMCA-P for the investigation of which classification of metabolites closely associated with the evolution of *Salvia* (Supplementary Data 2–7). ATDs accumulation in roots and leaves led to discernible division among the four clades using principal component analysis (PCA) when compared with phenolic acids, particularly the clustering results in the positive ion mode (Supplementary Figs. 25 and 26), which was consistent with the phylogenetic analysis results (Fig. 1).

In Fig. 2a, the 36 most prevalent ATDs were identified in both roots and leaves from the 71 *Salvia* species and were classified into five groups (A–E) based on structural characteristics. Reference standards and diagnostic neutral losses were used to identify the chemical modifications of rings and methyl groups at C-20 (Supplementary Data 8 and Figs. 27–31). Figure 2b depicts the accumulation patterns of the 36 ATDs in the 71 *Salvia* species, as well as the relationships between chemical modifications and organ specificity. Hydroxylation and carbonylation on the rings were the common reactions in the ATD biosynthetic pathways that occur in all four clades of *Salvia*. The accumulation of 20-keto and 20-carboxyl ATDs displayed species specificity in Clade I, II, and PR. Furthermore, strong organ specificity found in carboxylated and epoxide derivatives, such as carnosic acid and carnosol, were detected only in leaves of individual species from Clade I, II, and PR. In contrast to other ATDs with 20 carbon atoms, tanshinones (group B) had a greater ionic response in the positive ion mode and specifically accumulated in roots. Thus, although ATD chemical diversity was derived from various chemical modifications of rings and methyl groups, the accumulation pattern and organ specificity were correlated with specific methyl group modifications. Miltirone (Compound 6 in Fig. 2), the key precursor of tanshinone synthesis, was present in all clades and is formed by the unclear reactions of coupled demethylating C-20 and aromatizing ring B based on 11-hydroxyferruginol[15]. However, more-diverse tanshinones formed by subsequent heterocyclization, demethylation, and aromatization are present only in Clade IV[15]. Thus, C-20 is a potentially critical position: when the rings and methyl groups at C-20 are modified by varied substitutions, it correlates with the formation of diverse ATD characteristic structures and promotes species and organ specificity in *Salvia*.

## Catalytic divergence of CYP76AK subfamily

As the oxidation pattern on C-20 might bring the divergence of ATDs across the *Salvia* lineages. Thus, CYP76AK, a Lamiaceae-specific CYP subfamily that was previously discovered to catalyze the C-20 oxidation[27–30,32], might have shaped such metabolic diversity. To examine the enzyme activity across the CYP76AKs, all *CYP76AK* genes were annotated from the 71 transcriptomes we generated to study their phylogenetic relationship first. A total of 185 putative full-length sequences were obtained and used to generate a maximum likelihood (ML) tree (Fig. 3), which classified the CYP76AK subfamily into nine well-resolved clades (namely CYP76AK1, 2, 3, 5, 6, 7, 8, 18, and 22; Supplementary Data 9 and Fig. 32) with broadly taxonomic distribution. CYP76AK6s were unique to Clade I, as well as the only CYP76AK clade in this lineage. CYP76AK7, 22, 18 were specific to the Clade II lineage. Meanwhile, CYP76AK1, 2, and 3 were basically specific for the Clade IV lineage. Besides, CYP76AK3, 7, 8, and 22 genes were also annotated from the transcriptomes of PR.

Till now, only a few CYP76AK members have been functionally characterized[27–30]. To investigate whether each clade exhibits divergent catalytic properties, in total sixteen CYP76AKs, covering all clades, were selected for in vitro functional investigation using yeast microsome expression system. Two substrates, 11-hydroxyferruginol and 11-hydroxysugiol were tested to elucidate the oxidation patterns on C-20. Because 11-hydroxyferruginol was not commercially available, SmCYP76AH3 was introduced into the yeast strain WAT11 to transfer ferruginol into 11-hydroxyferruginol as the source of substrate (Fig. 4a). The sequential oxidation products were analyzed using LC-MS according to their MS features (Supplementary Fig. 33). When using 11-hydroxyferruginol as substrate, two products (11, 20-dihydroxyferruginol and carnosic acid) were produced, representing the products for hydroxylation and carboxylation on C-20. Among the CYP76AKs we tested, no products were observed for SmCYP76AK3 and 5. SmCYP76AK2, SpCYP76AK22, and ScCYP76AK18 exhibited hydroxylation activity, and only ScCYP76AK18 had subsequent carboxylation activity. Taken together with the previously functional descriptions on CYP76AK1, 6, 7, and 8[27–30], we assume that CYP76AKs exhibit three distinct catalytic properties towards 11-hydroxyferruginol, i. e., CYP76AK1, 2, and 22 can only exhibit hydroxylation of C-20, whereas CYP76AK6, 7, 8, and 18 had the further oxidation properties to produce carnosic acid. Besides 11-hydroxyferruginol, SmCYP76AK1 was able to convert 11-hydroxysugiol into 11, 20-dihydroxysugiol[27]. To test if other CYP76AK clades have promiscuities as well, 11-hydroxysugiol was fed into the microsomes preparations expressing each selected CYP76AK (Fig. 4b). As a result, all CYP76AKs except CYP76AK3 and 5 were capable of catalyzed C-20 oxidation of 11-hydroxysugiol. Different from using 11-hydroxyferruginol as substrate, 20-keto products were detected in the products of CYP76AK6, 7, 8, and 18. In addition, carboxylated products (7-keto-carnosic acid was purified and identified by nuclear magnetic resonance (NMR), Supplementary Fig. 34) were obtained in turn. Consistent with the 11-hydroxyferruginol feeding groups, CYP76AK1, 2, and 22 only exhibited one-step oxidation to produce 11, 20-dihydroxysugiol. In addition, we also selected complementary CYP76AKs from each clade for confirmation of function (Supplementary Fig. 35), demonstrating that CYP76AKs from the same clade had consistent functions.

Collectively, CYP76AKs appear to perform three types of catalytic properties, while adhering distinct phylogenetic divergence (Fig. 4c): i: CYP76AK7, 6, 18, and 8 were able to catalytic successive oxidation on C-20; ii: CYP76AK1, 2, and 22 only exhibited the hydroxylation step; iii: CYP76AK3 and 5 showed no catalytic ability to both 11-hydroxyferruginol and 11-hydroxysugiol. To further take comprehensive consideration of the catalytic properties, phylogenetic relationship of CYP76AK subfamily, and the ATDs distribution, there appears to be a strong correlation between the function divergence of

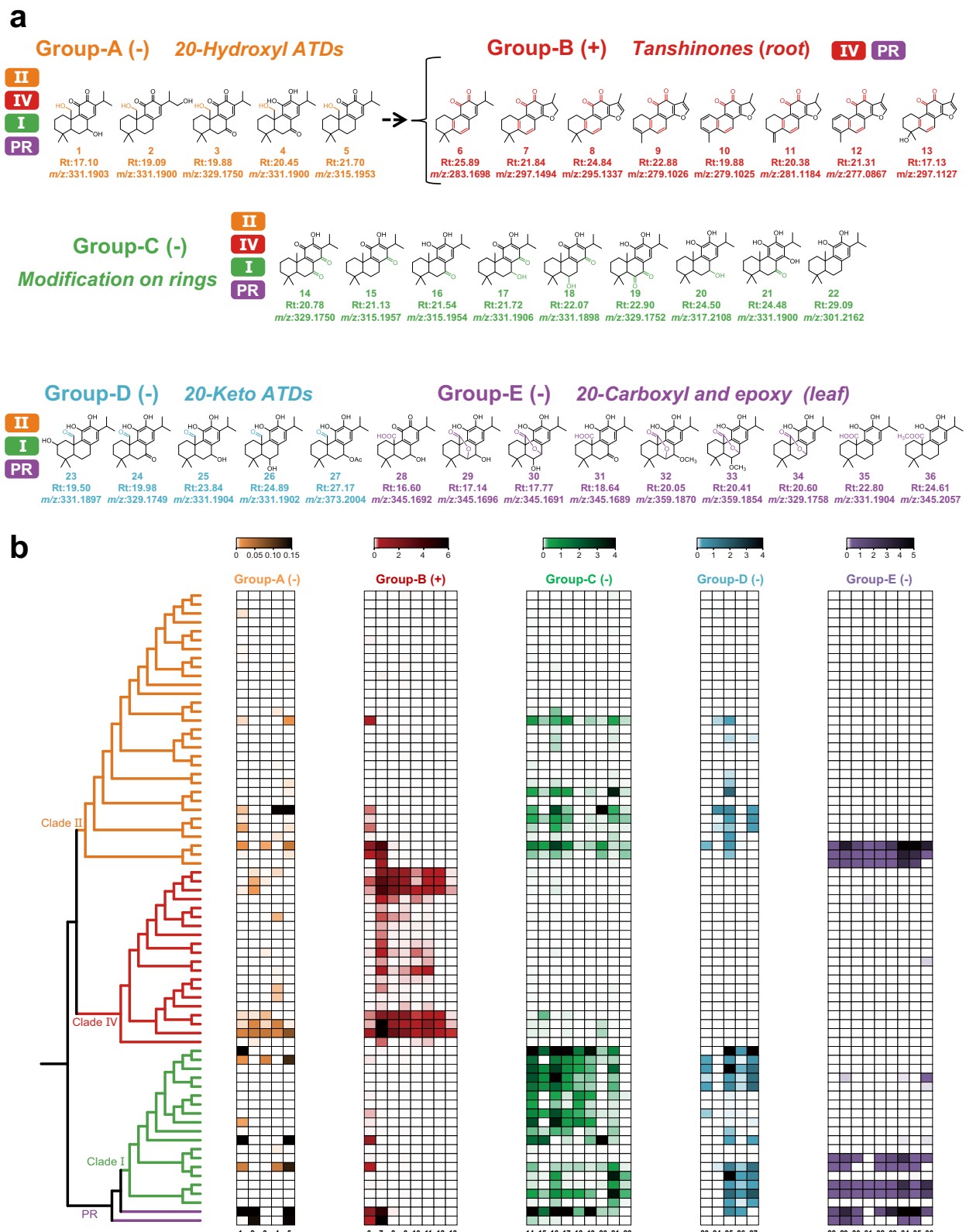

CYP76AKs and species-specific accumulation of ATDs among different *Salvia* lineages. For instance, 20-keto products as 20-keto-11-hydroxysugiol were specifically generated by CYP76AK6, 7, 8, and 18. Correspondingly, such 20-keto ATDs were restricted to Clade I, II, in addition to the PR (Fig. 2), which were consistent with the distribution of CYP76AK clades, i.e., CYP76AK6 might contribute for Clade I, CYP76AK7 and 8 in Clade II of the genus.

Organ constraint transcriptional patterns of CYP76AK genes provided further evidence for their contributions to ATDs diversity. Using RNA sequencing, we tested transcription patterns of CYP76AK genes in leaves and roots from seventeen selected *Salvia* species (Fig. 5), revealing their distribution according to lineages and diverse organ-specific expression patterns. Generally, *CYP76AK* genes belonging to the same clade had identical transcription pattern, and

**Fig. 2 | Chemical diversity and distribution of ATDs in *Salvia*.** The most prevalent ATDs identified by LC-MS in both roots and leaves from *Salvia*. **a** Thirty-six ATDs were divided into five groups based on different structural characteristics. ATDs in groups A, C, D, and E (which accumulated specifically in leaves) exhibited higher intensities in the negative ion mode, and ATDs in group B (which accumulated specifically in roots) exhibited higher intensities in the positive ion mode. **1:** 7, 20-Dihydroxyabietaquinone; **2:** 16, 20-Dihydroxyabietaquinone; **3:** 7-Keto, 20-hydroxyabietaquinone; **4:** 11, 20-Dihydroxysugiol; **5:** 20-Hydroxyabietaquinone; **6:** Miltirone; **7:** Cryptotanshinone; **8:** Tanshinone IIA; **9:** 1, 2-Dihydrotanshinquinone; **10:** Dihydrotanshinone I; **11:** Methylenetanshinquinone; **12:** Tanshinone I; **13:**

Przewaquinone C; **14:** 7-Keto-royleanone; **15:** Royleanone; **16:** 11-Hydroxysugiol; **17:** 7-Hydroxyroyleanone; **18:** 6-Hydroxyroyleanone; **19:** 6-Keto-11-hydroxysugiol; **20:** 7, 11-Dihydroxyferruginol; **21:** 11, 14-Dihydroxysugiol; **22:** 11-Hydroxyferruginol; **23:** 20-Keto-2, 11-dihydroxyferruginol; **24:** 20-Keto-11-hydroxysugiol; **25:** 20-Keto-7-hydroxyferruginol; **26:** 20-Keto-6-hydroxyferruginol; **27:** 20-Keto-7-acetyferruginol; **28:** 7-Hydroxy-20-carboxyabietaquinone; **29:** 7-Hydroxyisocarnosol; **30:** 6-Hydroxycarnosol; **31:** 7-Keto-carnosic acid; **32:** 7-Methoxycarnosol; **33:** 6-Methoxycarnosol; **34:** Carnosol; **35:** Carnosic acid; **36:** Methyl carnosate.
**b** Distribution patterns of the 36 ATDs in 71 *Salvia* species. PR, subgenera *Perovskia* and *Rosmarinus*. Source data are provided as a Source Data file.

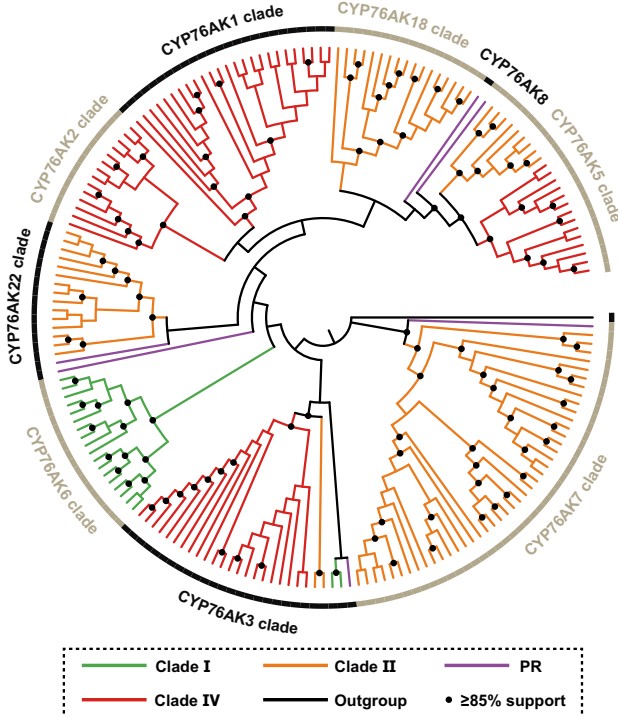

**Fig. 3 | Cladogram of the CYP76AK subfamily members.** The branch color represents the lineage to which the species containing these CYP76AK genes belong. See Supplementary Fig. 32 for the annotated phylogram. PR, subgenera *Perovskia* and *Rosmarinus*. Source data are provided as a Source Data file.

presented identical organ distribution to corresponding metabolites in the lineage. For instance, in Clade IV and Clade II, respectively, CYP76AK1 and 22 exclusively catalyzed the C-20 hydroxylation with root-specific expression, demonstrating their distinct role in the production of tanshinones. In contrast, 20-carboxyl products such as carnosic acid were specifically accumulated in the leaves in the basal lineages of Clade I and Clade II, respectively, which were most likely due to the organ-specific expression of CYP76AK6, 7, 8, and 18 in leaves of corresponding *Salvia* species. It should be noted that although CYP76AK3 and 5 did not demonstrate activity towards the tested substrates in our study, substantial transcription levels were found in the roots of the relevant *Salvia* species, indicating their unrevealed catalytic functions towards other unidentified substrates.

### Evolution of the ATDs biosynthetic pathway in *Salvia*
The above phylogenetic, metabolic, and catalytic functional studies revealed that this distinct metabolic divergence was accompanied by the evolution of the CYP76AK subfamily in each *Salvia* lineage. To trace the evolutionary route of the ATDs pathway in *Salvia*, comparative genome analysis and reconstruction of ancestral genes and metabolic traits were performed. Currently, there are six *Salvia*

genomes available, enable us to investigate the evolutionary history of *CYP76AK* genes in the majority of *Salvia* lineages. These are *S. rosmarinus*[50] (PR), *S. officinalis*[30] (Clade I), *S. miltiorrhiza*[46] and *Salvia bowleyana* Dunn[51] (Clade IV), *S. hispanica*[52] (North American lineage of Clade II), and *S. splendens*[37] (South American lineage of Clade II). First, microsynteny blocks were investigated to determine the evolutionary history of CYP76AKs (Fig. 6a). Conserved syntenic regions containing CYP76AKs were observed in all studied genomes and were not located in the known gene cluster related to diterpenoid biosynthesis[30,53]. Two duplicated regions were present in the *N. cataria* genome, allowing the tracing of the ancestor of CYP76AK genes before the speciation of *Salvia*. Meanwhile, duplication of the region was discovered that showed an obvious correlation with the WGD event of each genome. For *N. cataria* and *S. rosmarinus*, the duplication was probably produced by their unique WGD events. In the *S. hispanica* genome, the duplication was related to the Clade II-common WGD. Subsequently, *S. splendens* experienced another unique WGD event, which eventually resulted in a genome with four duplicated regions. Thus, the discovery of the syntenic regions and their duplication events suggested that this region contained ancestral CYP76AKs in the genome of the *Salvia* ancestor. Along with genome evolution, CYP76AK originated and evolved into divergent clades in different *Salvia* lineages. In the phylogeny of both CYP76AKs (Fig. 3) and *Salvia*, CYP76AK7 and 22 might have emerged first in the ancestor of the *Salvia* lineages as shown in the *S. rosmarinus* genome, and subsequent duplication events then led to CYP76AK8 (*S. rosmarinus*) and CYP76AK18 (Clade II). In taxa that have not undergone WGD events, CYP76AK is separated into CYP76AK1 (Clade IV) and CYP76AK6 (Clade I). In addition, no collinearity relationship for CYP76AK2, 3, and 5 were discovered among all *Salvia* genomes (Supplementary Data 10). Their emergence might have been caused by unique duplication events in certain species or lineages (e.g., segmental and transposon-mediated duplication)[54].

In consideration of the functional differences among CYP76AKs within the context of their evolutionary histories, the divergence of catalytic activity at C-20 seemed to represent a taxonomic loss of function rather than a neofunctionalization. Thus, ancestral gene reconstruction and function validation were performed. Based on the phylogeny of CYP76AK, six branch nodes were selected for ancestral enzyme resurrection (Fig. 6b, Supplementary Data 11). The ancestral enzyme corresponding to each of these nodes was expressed in yeast and functionally confirmed using 11-hydroxyferruginol and 11-hydroxysugiol as the substrates. Consistent with expectations, all CYP76AK ancestors exhibited hydroxylation, carbonylation, and carboxylation at C-20 of both 11-hydroxyferruginol and 11-hydroxysugiol, as each oxidation product was produced by each CYP76AK ancestor (Fig. 6c, d). Node 6 represented the most ancestral CYP76AK protein, which exhibited a complete oxidation function, indicating that such catalytic trait of CYP76AKs had already been acquired when the CYP76AK subfamily initially emerged in *Salvia*. Node 3 (ancestor of CYP76AK1, 2, 5, 8, 18, and 22) and Node 2 (ancestor of CYP76AK1, 2, 5, 8, and 18) also exhibited complete oxidation capacities toward C-20, implying that CYP76AK1, 2, and 22 (only performing hydroxylation) lose the carboxylation activity. CYP76AK3 and 5 showed no catalytic

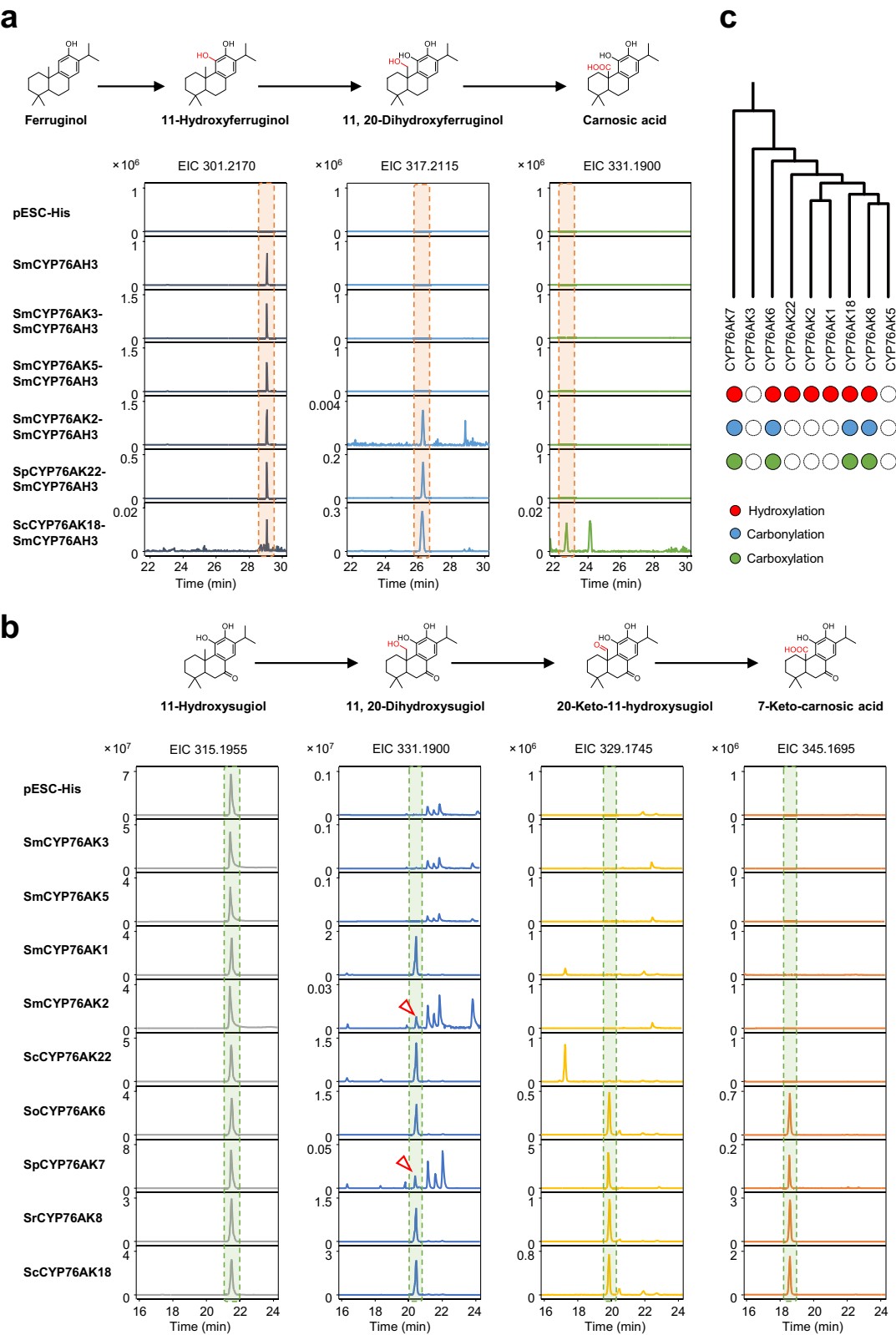

**Fig. 4 | Catalytic activity of targeted CYP76AK subfamily members with two substrates. a** Extracted ion chromatogram (EIC) analysis of the CYP76AK catalytic reaction products in vitro with 11-hydroxyferruginol. To obtain 11-hydroxyferruginol, all plasmids contained SmCYP76AH3, which converts ferruginol to 11-hydroxyferruginol. A sample with only SmCYP76AH3 protein functioned as the negative control. Peaks corresponding to potential products are indicated by the m/

z value of 11, 20-dihydroxyferruginol ($m/z$ 317.2115), and carnosic acid ($m/z$ 331.1900). **b** EIC overlays showing in vitro catalytic activity of CYP76AKs with 11-hydroxysugiol. The sample with the empty vector (pESC-His) functioned as the negative control. Peaks corresponding to potential products are indicated by the $m/z$ value of 11, 20-dihydroxysugiol ($m/z$ 331.1900), 20-keto-11-hydroxysugiol ($m/z$ 329.1745), and 7-keto-carnosic acid ($m/z$ 345.1695). **c** Enzyme activity summary of CYP76AKs.

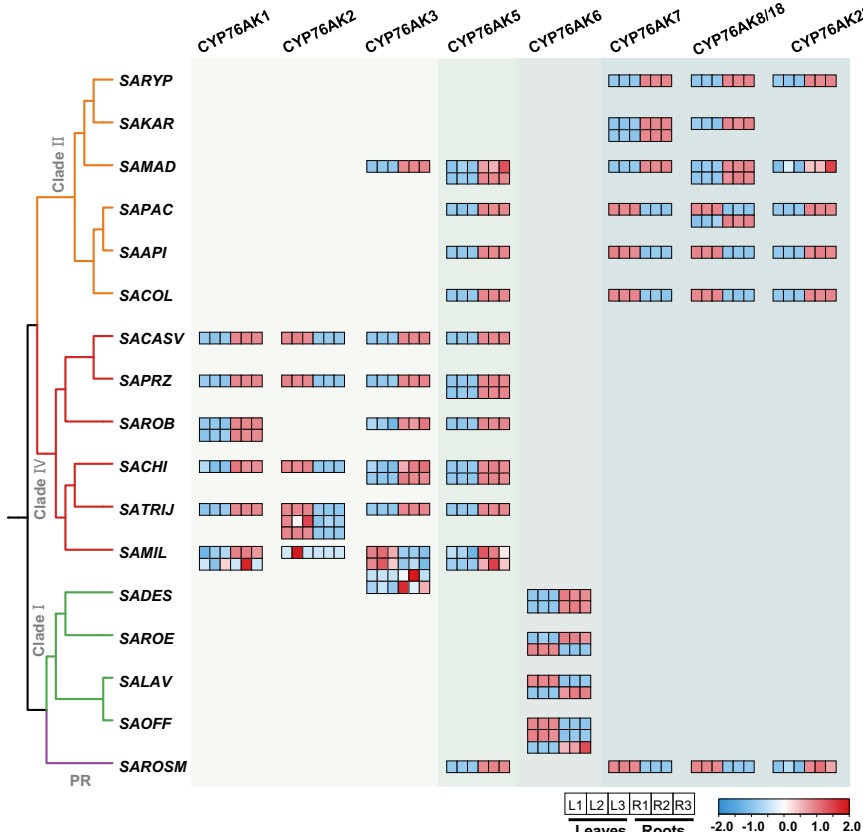

**Fig. 5 | Expression profiles of the CYP76AK subfamily members in *Salvia*.** Gene expression heatmap for CYP76AK subfamily members from representative species of each branch of Salvia, displaying the relative expression levels in leaves (L) and roots (R). The heatmap shows three biological replicates, each plotted individually. Expression values represent *Z* score-transformed log (TPM + 1) values. Species abbreviations can be found in Supplementary Data 1. Source data are provided as a Source Data file.

activity with any of the substrates assessed (Fig. 4), suggesting that CYPs in the two clades experienced complete loss of activity towards the tested substrates. They might undergo pseudogenization or have catalytic activities towards other unknown substrates.

Considering the evolutionary history of CYP76AKs in *Salvia*, the metabolic diversity of ATDs appears to represent a pathway loss rather than metabolic innovation. Thus, ancestral phenotype reconstruction (APR) was performed according to the realistic chemical traits in each taxon to predict the chemical traits of their common ancestors (Supplementary Fig. 36). The abilities to produce carnosic acid, tanshinone, or 20-keto ATDs were considered as key traits. APRs for the early speciation nodes of the *Salvia* phylogeny (PR and Clade I, II, and IV) were indicated to produce all types of ATDs, implying that the ancestor of *Salvia* most likely produced all ATDs. With the speciation of *Salvia*, this chemical trait was retained in PR (*S. rosmarinus*) and the North American taxa in the Clade II lineage, but it was partially lost in other lineages. Loss of activity at the clade level was observed in the evolutionary history of CYP76AKs, thus suggesting a major cause for the metabolic diversity among lineages. To model how the evolution of activity of CYP76AKs directed ATDs diversity, the chronology of all CYP76AKs was further assessed (Supplementary Fig. 37). The CYP76AK clades that originated from the ancestral syntenic segments were presented (Fig. 6e, f). The chronology revealed that the MRCA of CYP76AK (-76.4 Mya) was present at a time much earlier than the first divergence between CYP76AK7 and the other clades (-45.1 Mya). Each clade emerged at the time of, or just after, the speciation of its corresponding lineages. For example, CYP76AK6 might have emerged along with the speciation of Clade I (-20.4 Mya). Moreover, the loss of activity of CYP76AK22 (-26.3 Mya) and CYP76AK1 (-7.9 Mya) occurred

after the speciation of *S. rosmarinus* (-27.2 Mya) and Clade IV (-9.1 Mya), respectively.

Based on the catalytic traits, lineage distribution, and transcriptional pattern of each CYP76AK clade, the gene contributing to ATDs biosynthesis in a particular lineage can be determined. CYP76AK7 was found in *S. rosmarinus* and Clade II, thus contributing to carnosic acid biosynthesis in the aerial parts of the corresponding lineage. However, CYP76AK7 might also be responsible for the accumulation of 20-keto ATDs in roots of species in the South American taxa of Clade II. CYP76AK6 was present only in Clade I; thus, it might enable the biosynthesis of carnosic acid-related metabolites and 20-keto ATDs in specific organs. CYP76AK22 was retained in *S. rosmarinus* and Clade II and is expressed only in roots where it is responsible for tanshinone accumulation. Meanwhile, CYP76AK1 contributed to tanshinone biosynthesis only in Clade IV. In addition, CYP76AK8/18 shared a similar distribution to CYP76AK22 and might perform hydroxylation and carbonylation in roots for the biosynthesis of 20-keto ATDs. Overall, previous studies on the metabolic evolution of ATDs and our findings concerning the functional divergence of the CYP76AK subfamily support a model of metabolic diversity in *Salvia* lineages based on loss of activity.

## Discussion

In this study, we combined lineage-wide phylogeny, metabolic profiling, genomic comparison, and enzyme function evolution analyses to illustrate how ATDs divergence in the genus *Salvia*. Our metabolic profiling for ATDs provides the model for ATDs pathway evolution and serves as an example of chemical diversity in plants due to the loss of activities of catalytic enzymes.

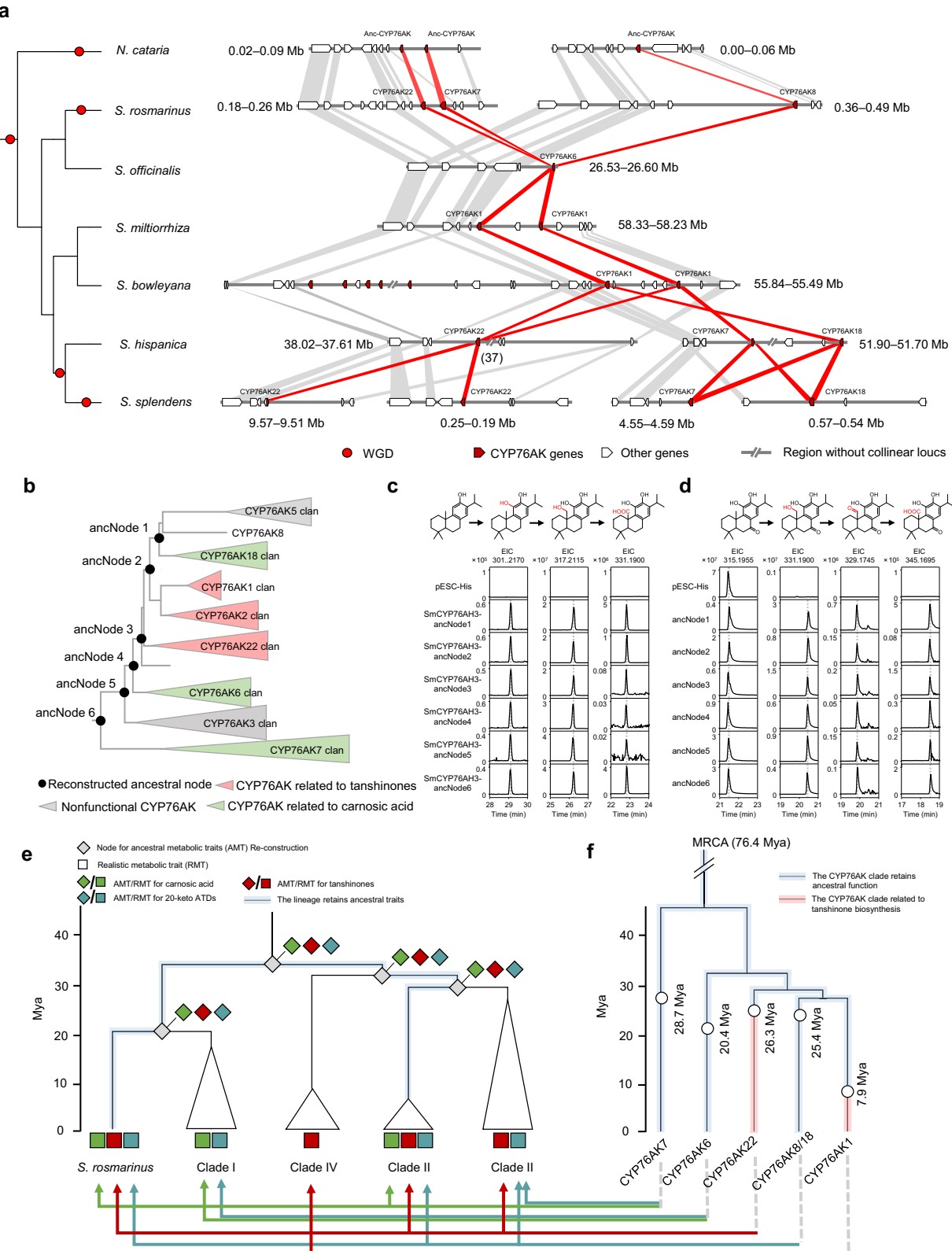

Previous phylogenetic reconstructions of *Salvia* were mainly based on ribosomal, mitochondrial, and chloroplast genes, with a few protein-coding nuclear genes having been used in the phylogeny of *Salvia* at the family level[47,48,55–60]. Uniparental heritability of organellar genes, as well as gene recombination and transformation in the plastid genome, has resulted in phylogenetic reconstruction biases and inaccuracies[61]. In contrast, nuclear genes have many advantages, such

as their large numbers and parental inheritance. The use of the thoroughly evaluated nuclear gene sets can provide evidence for robust and well-supported deep angiosperm lineages[62–65]. Here, we presented phylogenetic analysis of *Salvia* based on large-scale nuclear genes, obtained from transcriptome sequencing with the largest sampling scale for *Salvia* lineage to date, to infer infrageneric relationships within the genus. Consistent with previous studies, *S. abrotanoides* and

**Fig. 6 | Evolution of the CYP76AK subfamily. a** Syntenic relationships among *CYP76AK* genes in *Salvia* genomes revealed the divergence of CYP76AK clades. The phylogenetic tree in the left panel represents the phylogeny of six *Salvia* species with *N. cataria* as the outgroup. WGD events are indicated by red dots. Micro-synteny of *CYP76AK* genes in the synteny blocks. Red polygons show syntenic relationships between *CYP76AK* genes, and gray polygons show other genes. Anc-CYP76AK, Ancestral *CYP76AK* gene. **b** Reconstruction of ancestral CYP76AK proteins. Ancestral nodes (ancNodes) 1–6 show the resurrected enzyme of each major branch point of the CYP76AK phylogenetic tree. Functional characters of each branch are distinguished by different colors. **c, d** Ancestral function of CYP76AKs. Extracted ion chromatograms (EICs) showing in vitro catalytic activity of each ancestral CYP76AK using 11-hydroxyferruginol and 11-hydroxysugiol as the substrate, respectively. **e** Evolution of ATDs biosynthesis in *Salvia*. The phylogenetic tree shows the speciation of the clades of *Salvia*, and the diamonds represent the predicted ancestral metabolic traits for each speciation node. Square frames represent the realistic metabolic traits of each *Salvia* clade as shown in Fig. 3. Green, red, and light blue represent the existence of carnosic acid, tanshinones, and 20-keto ATDs, respectively. The time scale on the left reflects the time of speciation for each clade. Branches highlighted in light blue indicate clades in which the ancestral metabolic traits have been retained. **f** Chronology of *CYP76AK* genes. The phylo-genetic tree represents the evolution of CYP76AK clades according to the MCMCtree model. The speciation time for each clade is labeled. Branches high-lighted in blue indicate CYP76AK clades that have retained their ancestral catalytic functions, and branches highlighted in red indicate clades that have undergone a loss of function. The arrows at the bottom indicate the CYP76AK clade that con-tributes to producing the type of ATDs in the corresponding lineage. Green, red, and indigo gray arrows indicate the biosynthesis of carnosic acid, tanshinones, and 20-keto ATDs, respectively. Source data are provided as a Source Data file.

*S. rosmarinus* formed a clade with *Salvia* species, whereas *M. officinalis* acted as a sister clade[47,48,55,56,59,60].

Morphologically, the traditionally defined genus *Salvia* is distinct from other genera of Lamiaceae in that two of its fertile stamens are separated by a significantly elongated connective tissue. However, molecular phylogenetic studies expanded *Salvia* to include other genera (i.e., *Dorystaechas*, *Meriandra*, *Perovskia*, *Rosmarinus*, and *Zhumeria*)[47,48,55–60]. Here, although representatives of "*Rosmarinus*" and "*Perovskia*" formed a sister group to Clade I (Fig. 1), the metabolic traits for ATDs and catalytic properties of CYP76AKs in two subgenera are distinctly different from that of Clade I. Thus, the above findings could provide new insights from more aspects into the taxonomic attribution of *Dorystaechas*, *Meriandra*, and *Zhumeria*.

CYPs oxidize the hydrocarbon skeleton generated by terpenoid synthase during plant terpenoid biosynthesis, while chemical mod-ifications include hydroxylation, continuous oxidation, ring rearran-gement, and heterocyclization also occur. These reactions greatly increase the structural diversity of terpenoids, while providing anchor points for future moiety modifications. Therefore, CYPs are con-sidered the key factors for the diversity of plant terpenoids[32]. According to our metabolome analyses, ATDs with modified rings (group C) showed no specific accumulation in all clades and were produced by functional genes such as CYP76AH subfamily, which is able to carry out carbonylation at C-7 of ring B[27]. In contrast to the modifications carried out by the CYP76AH, the chemical modifications at C-20 are driven by the diverse CYP76AK family, which is responsible for the accumulation and organ specificity of ATDs in *Salvia*. Accord-ing to Figs. 2 and 5, the *CYP76AK1* gene was highly expressed in roots of taxa in Clade IV and produces 20-hydroxyl ATDs (group A), the accu-mulation patterns of which were similar to those of miltirone, especi-ally for the 7, 20-dihydroxyabietaquinone (compound 1), 7-keto-20-hydroxyabietaquinone (compound 3), and 20-hydroxyabietaquinone (compound 5), which is speculated that the 20-hydroxyl ATDs are the primary precursors for miltirone synthesis. Being similar to CYP76AK1, CYP76AK22 was also found to produce hydroxylated derivatives at C-20 for miltirone synthesis in individual species from Clade II and PR, and had high transcriptional level in roots as well (Fig. 5). Contrary to Clade I, species fall into Clade II as *S. pachyphylla*, *S. apiana*, and *S. columbariae* produce cryptotanshinone. The production of crypto-tanshinone (Compound 7; Fig. 2) by the CYP71D family acting on mil-tirone for the heterocyclization of ring D showed that Clade II species had a more comprehensive tanshinone biosynthetic pathway than species in Clade I (Fig. 2)[53]. However, most downstream tanshinones were more abundant in Clade IV, indicating that ATDs biosynthesis in Clade IV consisted predominantly of tanshinone synthesis (group B). This was also the possible reason why the common ATDs with 20 carbon atoms (group C) in Clade I and II accumulated to higher levels (Fig. 2). In addition, the subsequent aromatization of ring A requires functioning genes for hydroxylation at C-18 or −19, which were exclusively expressed in Clade IV. Besides, a side product of

SmCYP76AK1 and ScCYP76AK22 was detected when 11-hydroxysugiol was used as the substrate (time = 17.25 min; Fig. 4b), which was tenta-tively identified as 6-hydroxy-7-ketoabietaquinone (Supplementary Fig. 38). This side function might contribute to the ATDs with hydroxyl group at C-6 in Clade II (Compound 18) and IV (Compound 18, 26, and 30). Correspondingly, the enzymes response for C-6 oxidation in Clade I and PR still need to be explored.

Apart from the significance of CYP76AK1 and CYP76AK22 for the synthesis of 20-hydroxyl ATDs and tanshinones, the unique accumu-lation of 20-keto ATDs (group D) in Clade I, II, and PR was based on the specific expression of CYP76AK6, 7, and 8/18. Whereas CYP76AK6 was uniquely expressed in Clade I, CYP76AK7 and 8/18 were expressed in Clade II and PR, and performed similar activities of hydroxylation, carbonylation, and carboxylation at C-20 in the in vitro enzyme activity assays (Fig. 4). In the enzyme activity assays, the 20-carboxyl deriva-tive**s**, such as carnosic acid (Compound 35), were also found to spontaneously oxidize 20-epoxy ATDs such as carnosol (Compound 34) (Supplementary Fig. 39)[28]. However, species with higher expres-sion of CYP76AK6, 7, or 8/18 in leaves, such as *S. officinalis* in Clade I, *S. columbariae* in Clade II, and *S. rosmarinus* in PR, were the few species with both 20-carboxyl and 20-epoxy ATDs (group E) in leaves (Figs. 2 and 5). Such species-specific accumulation of carnosic acid–related metabolites may be related to the ability to withstand the harsh cli-matic conditions in the Mediterranean habitat[66]. Leaves from these species showed higher expression of CYP76AKs than roots, which might be related to the specific localization of 20-carboxyl and 20-epoxy ATDs in the chloroplasts of photosynthetic green organs[67,68]. Carnosic acid was previously speculated to be a potential intermediate for demethylation (C-20), leading to the formation of a heterocyclic bridge between C-20 and C-7. This, in turn, suggested that the car-boxylic acid group on the C-20 may be responsible for aromatization of ring B[15], ultimately resulting in the production of miltirone. How-ever, the accumulation of carnosic acid was strict to certain organs and species, this indicates that it cannot serve as a precursor for tan-shinones. Instead, hydroxylation (C-20) catalyzed by CYP76AK1 and 22 demonstrates a stronger correlation with subsequent demethylation and aromatization due to its co-existence and similar accumulation pattern with tanshinones in Clade IV. This represents a significant step in the mechanistic investigation of the formation process of tan-shinones, specifically in understanding the mechanisms of demethy-lation and aromatization. Thus, the variable expression of distinct CYP76AKs between evolutionary clades might direct the evolution of ATD biosynthesis pathway, resulting in the specific production of ATDs in *Salvia*, such as tanshinones and carnosic acid-related metabolites.

In addition, based solely on the in vitro activity evidence of CYP76AK, we cannot rule out the possibility of other enzymes parti-cipating in the consecutive oxidation of ATDs. Similar to the artemi-sinin biosynthetic pathway, although in vitro evidence shows that CYP71AV1 can catalyze the consecutive oxidation of amorpha-4,−11-

diene on C-12 to form artemisinic aldehyde and artemisinic acid, other enzymes such as alcohol dehydrogenase (ADH1) and artemisinic aldehyde dehydrogenase (ALDH1) are also involved in the last two oxidation steps in glandular trichome cells of *Artemisia annua*[69–71]. This also suggests the possibility of the involvement of other enzymes, in addition to CYP76AK, in the C-20 oxidation of ATDs in *Salvia*, which requires further in vivo functional studies of CYP76AKs and the discovery of new enzymes.

It is the activity loss of CYP76AKs rather than neofunctionalization that has driven the diversity of ATDs in the genus *Salvia*. The differentiation and expansion of CYP76AKs caused by WGD events have not resulted in neofunctionalizations as in previous cases[72] but rather in a loss of activities in corresponding taxa (Fig. 6). It is interesting to note that the loss of activity has also triggered changes in structural characteristics and distribution patterns that result in metabolic diversity, such as the production of tanshinones and carnosic acid derivatives. Contrary to most cases, which have assumed that functional innovation is what leads to metabolic diversity, the results from this study suggest that the loss of activities may also act as the key factor for metabolic diversity.

On the other hand, with respect to the functional divergence, CYP76AKs also exhibit distinct transcriptional patterns, which eventually lead to the completely opposite organ-specific accumulation of carnosic acid derivatives and tanshinones. On the basis of this, we speculate that the physiological and ecological functions of these metabolites may be connected to the metabolic divergence present in this genus. For instance, carnosic acid and carnosol are mainly accumulated in the chloroplasts of green tissues, thereby providing a unique and effective antioxidant mechanism in the photosynthetic tissues of these plants[66]. The carnosic acid-based protection mechanism is proved to be crucial for enabling *Salvia* species to withstand harsh climatic conditions, whereas tanshinones may be involved in the interaction between roots and the microorganisms in the surrounding soil[73,74].

Furthermore, the distinct transcription patterns of each CYP76AK clade represented another aspect of function evolution. Their organ-specific patterns are not independent, but are accompanied by other catalytic genes involved in carnosic acid derivatives and tanshinones biosynthesis as *KSL*s, *CYP76AH*s, and *CYP71D*s. In the meantime, such patterns are maintained in different *Salvia* lineages[30,53,75], thus, raising the question about the origin of their organ-specific transcriptions as a further research topic. To summarize, ATD differentiation in *Salvia* provides a complete model involving multiple species, gene families, metabolic diversity, and ecological evolution, making it an ideal research object for the genetic mechanisms responsible for chemical diversity in plants.

## Methods

### Plant materials

Most recent molecular phylogenetic studies support the presence of 11 subclades in the genus *Salvia* (subg. *Audibertia*, subg. *Calosphace*, subg. *Dorystaechas*, subg. *Glutinaria*, subg. *Heterosphace*, subg. *Meriandra*, subg. *Perovskia*, subg. *Rosmarinus*, subg. *Salvia*, subg. *Sclarea*, and subg. *Zhumeria*)[56,58–60]. A phylogenetic tree of *Salvia* was constructed to obtain a sampling scheme to analyze chemical diversity among these species (Supplementary Data 1). In total, 71 species from three major distribution centers for this genus representing 8 out of the 11 recognized subclades of *Salvia* were sampled. Six species (*M. officinalis*, *M. spicata*, *C. polycephalum*, *O. vulgare*, *N. cataria*, and *P. vulgaris*) from other Mentheae tribes were selected as outgroups. To obtain sufficient materials for both transcriptome and chemical analyses, samples were collected at the vegetative or flowering stages. Three organs (roots, stems, and leaves) were collected at the vegetative stage and four organs (roots, stems, leaves, and flowers) at the flowering stage. To permit sampling of the same tissue type for

transcriptome and chemical analyses, tissues from each organ were ground into a fine powder and divided into aliquots.

### RNA isolation, transcriptome sequencing, assembly, annotation, and analysis

In order to obtain more comprehensive transcripts, a mixed organ strategy for library construction was performed, by which the transcript database generated was used for phylogenetic studies. For transcription profiling, libraries for individual leaves and roots from representative plants in each clade were generated (i.e., *S. rypara*, *S. karwinskii*, *S. madrensis*, *S. pachyphylla*, *S. apiana* and *S. columbariae* from Clade II; *S. castanea f. tomentosa*, *S. przewalskii*, *S. roborowskii*, *S. chinensis*, *S. trijuga*, *S. miltiorrhiza* from Clade IV; *S. deserta*, *S. roemeriana*, *S. officinalis* subsp. *lavandulifolia*, *S. officinalis* from Clade I; *S. rosmarinus* from PR)[47,48]. Total RNA was isolated using the TransZol Plus RNA Kit (TransGen Biotech, Beijing, China). The cDNA libraries were prepared using the Illumina TruSeq RNA Sample Preparation Kit (Illumina, San Diego, CA, USA) and were sequenced on an Illumina NovaSeq 6000 sequencer, generating $2 \times 150$-bp paired-end reads. Reads were assessed and trimmed by fastp (https://github.com/OpenGene/fastp) with default parameters. The transcript assemblies were generated using Trinity assembler[76]. For the transcriptome annotation, all transcripts were searched against the *Arabidopsis thaliana* proteome, SwissProt, and Pfam databases using BLASTX, BLASTP, and HMMER v3.1b2 with an e-value cutoff of 1E-5, respectively[77]. The gene expression values (transcripts per million reads [TPM]) were calculated by RSEM software[78]. Essentially, differential expression analysis was performed using DESeq2[79] with *p* adjust <0.05 and |log2FC|≥1 as the threshold. The gene heatmaps were generated using TBtools (https://github.com/CJ-Chen/TBtools) with the Heatmap Illustrator function.

### Orthologous identification, single/low-copy gene selection, and phylogeny construction

All assembled transcripts were filtered to retain the longest transcript isoforms, and the protein sequences were further predicted using TransDecoder v5.5.0[80]. Orthologous groups were constructed with OrthoFinder v2.5.4[81] using TransDecoder-predicted protein sequences from all 77 species. Considering that fragmented assembly, incorrect assembly, insufficient informative sites, and other factors might result in biased inferences, the five gene sets including four low-copy gene sets, i.e., 2,178 OGs, 1,532 OGs, 1,169 OGs, and 512 OGs, and one single-copy gene set, i.e., 130 OGs, were extracted for the construction of phylogenetic tree. The resulting protein sequences from single/low-copy gene sets were aligned with MAFFT v7.487[82] using the default settings, and poorly aligned regions were trimmed by TrimAl v1.4. rev15[83]. The phylogenetic tree of each orthologous from individual OGs was constructed using RAxML-NG v1.0.3[84] with 100 replicates under the JTT + I + G4 model. Then, Astral v5.6[85] was used to concatenate the phylogeny trees for each of the five OG sets with 100 replicates from RAxML to obtain the BS values of all nodes. In addition, 130 OGs were concatenated into a supermatrix, and the final phylogenetic relationship was reconstructed using RAxML-NG v1.0.3[84] with 1000 replicates under the JTT + I + G4 model. Among all 77 species, *M. officinalis*, *M. spicata*, *C. polycephalum*, *O. vulgare*, *N. cataria*, and *P. vulgaris* were chosen as the outgroups.

### Divergence time estimation

MCMCTree in the PAML package v4.9[86] was employed for Bayesian inference of divergent time estimation of candidate species. The topology of candidate species from ML reconstruction with the concatenated 130 OGs was used as the input tree. The resulting nucleotide sequences with codon substitution models were aligned and trimmed with MAFFT v7.487[82] and the "backtrans" parameter of TrimAl v1.4. rev15[83]. Firstly, the branch lengths and overall substitution rate (rgene

gamma) were measured using the BASEML program in the PAML package v4.9[86] under the GTR + G model. Secondly, the divergent time was estimated using MCMCtree, and the parameters of burn-in, sampfreq, and nSample were set to 500,000, 150, and 10,000, respectively. The divergence time of candidate *Salvia* species was estimated based on the following fossil age constraints: 11-27 Mya for the divergent of *C. polycephalum* and *M. spicata*, and 21–47 Mya for the divergent between *M. officinalis* and the other outgroups, i.e., *N. cataria*, *C. polycephalum*, *M. spicata*, *O. vulgare*, and *P. vulgaris*. Lastly, multiple iterations of MCMCTree estimation made the deviation of divergent time <0.1%.

### WGD events identification

$K_S$-based age distributions for paralogs of all candidate species were constructed using the "wgd" pipeline[87]. Briefly, the paralogs for each species were identified using the Diamond v0.9.18.119 sequence similarity search tool with an E-value cutoff of 1E-10[88]. The paralogous gene families were clustered using the Markov cluster algorithm[89]. The genes within each paralogous gene family were aligned using MAFFT v7.487[82] with the default parameters, respectively. The $K_S$ values of all paralogous gene pairs within one gene family were measured using the CodeML program in the PAML package v4.9[86]. $K_S$ values were subsequently node-weighted to correct for the redundancy with the phylogenetic tree construction for each family using FastTree v2.1.7[90]. The $K_S$ aging distribution of paralogs of all tested species is shown in the gray bars of Supplementary Fig. 7.

Given the different substitution rates of all tested species, the species-shared or specific WGD events were further adjusted using a $K_S$-based tree. Here, 130 single-copy genes from OrthoFinder v2.5.4[81] for all tested species were used to calculate the branch lengths of phylogeny in the $K_S$ unit using the CodeML program in the PAML package v4.9[86] with a free-ratio model.

### CYP76AK sequences, phylogenetic analyses, and molecular dating analysis

The genes with the Pfam domain of PF00067 were identified as CYP450 family members. Seven known CYP76AK genes (*SmCYP76AK1*: KR140169.1, *S. miltiorrhiza*; *SmCYP76AK2*: KP337688.1, *S. miltiorrhiza*; *SmCYP76AK3*: KP337689.1, *S. miltiorrhiza*; *SfCYP76AK6*: KX431218.1, *S. fruticosa*; *SpCYP76AK6*: KT157045.1, *S. pomifera*; *SrCYP76AK7*: KX431219.1, *S. rosmarinus*; and *SrCYP76AK8*: KX431220.1, *S. rosmarinus*) were downloaded from the NCBI database as a local CYP76AK database. BLASTP searches were performed to identify the corresponding *CYP76AK* genes with an E-value cutoff of 1E-5. The protein sequences of the selected *CYP450* genes that encoded proteins of ≥466 amino acids were aligned and trimmed using MAFFT v7.487[82] and TrimAl v1.4.rev15[83], respectively. The phylogeny of *CYP76AK* genes was inferred using *CYP76AH1* gene as outgroup via ModelTest-NG v0.1.7[91] and RAxML-NG v1.0.376[84]. For *CYP450* genes that encoded proteins of <466 amino acids, BLASTP searches were used to classify the candidate genes.

According to the phylogenetic tree above, the molecular dating of CYP76AKs from different species was analyzed using MCMCTREE of the PAML package v4.9[86] under the GTR + G model (model = 7) with 500000 iterations and 150 sample frequencies, following 500,000 iterations as burn-in. The substitution rate, i.e., rgene gamma was calculated as G (1, 5.95) using BASEML of the PAML package v4.9[86]. The crown node of two single-copy gene families (i.e., CYP76AK3 and 22) and two species-specific gene families (i.e., CYP76AK1 and 6) were used to estimate the divergent time of each node. We used the following age constraints for each estimation procedure: a minimum and maximum age of 23.7 and 35.2 Mya for the crown node of CYP76AK3 (the divergence time of *S. abrotanoides* and other species); a minimum and maximum age of 21.8 and 32.9 Mya for the crown node of CYP76AK22 (the divergence time of *S. rosmarinus* and others); a minimum and

maximum age of 14.3 and 22.5 Mya for the crown node of CYP76AK6 (the divergence time of Clade I and others); a minimum and maximum age of 6.7 and 11.8 Mya for the crown node of CYP76AK1 (the divergence time of Clade IV and others).

### Collinearity analysis of CYP76AK members

Synteny blocks between each pair of candidate species (*N. cataria*, *S. rosmarinus*, *S. miltiorrhiza*, *S. bowleyana*, *S. hispanica*, and *S. splendens*) were performed using MCScan (Python version). The targeted CYP76AKs or neighboring genes were chosen as seeds to search for synteny blocks of conserved evolution.

### Reconstructions of ancestral sequences and traits

Using the matched sequence file, phylogenetic tree file, and configured control file, the JTT + GAMMA model of the CodeML program in the PAML package v4.9[86] was used to estimate the hypothetical ancestor sequence of CYP76AK subfamily members. The regions with alignment gaps were analyzed by the parsimony method to determine the ancestral residue base. The estimated ancestral genes were then synthesized and codon-optimized for functional identification in *S. cerevisiae*.

Based on the metabolic results and literature research[39], the species composition types of *Salvia* were coded. The component types are coded into 6 states: none (A); TAs (B); 20-keto ATDs (C); TAs and 20-keto ATDs (D); CAs and 20-keto ATDs (E); TAs, CAs, and 20-keto ATDs (F). Taking the evolutionary tree (ML) with branch length information as the input tree, Statistical Dispersal-Vicariance Analysis (S-DIVA) from Reconstruct Ancestral State in Phylogenies (RASP) software was used to infer ancestral states of component types and calculate phylogenetic signals[92].

### Chemical standards

Chemical standards were purchased from Shanghai Standard Technology Co., Ltd. (Shanghai, China). The other chemicals and reagents were purchased as follows: acetonitrile and methanol (HPLC grade; Merck, Darmstadt, Germany), warfarin (Sigma-Aldrich, Madrid, Spain), chloroform (Sinopharm Chemical Reagent, Shanghai, China), leucine encephalin (Waters, Milford, MA, USA), pure distilled water (Watsons Water, Hong Kong, China), and formic acid (HPLC grade; Fisher Scientific, Fairlawn, NJ, USA).

### Extraction and analysis of ATDs by UPLC-QTOF-MS

Each plant was separated into root and leaf samples, with three biological replicates. All samples were ground into a fine powder, and 10 mg prepared powder was extracted with 1 mL of 70% methanol (v/v) containing warfarin (5 μg/mL) as the internal reference for 1 h in an ultrasonic bath (53 kHz, 350 W) at 4 °C. After centrifuging at 12,000 × g at 4 °C for 30 min, the supernatant was used for UHPLC-QTOF-MS analysis.

An Acquity UPLC system (Waters, Milford, MA, USA) coupled with a Xevo G2-XS QTOF mass spectrometer (Waters, Milford, MA, USA) was used for the metabolic analysis. The samples were first separated using an Acquity UPLC T3 column (2.1 mm × 100 mm, 1.8 μm). The column temperature was kept constant at 40 °C, and the flow rate was 0.40 mL/min with an injection volume of 1.0 μL. The mobile phases for gradient elution consisted of 0.1% (v/v) formic acid/water (solvent A) and 0.1% (v/v) formic acid/acetonitrile (solvent B). The elution gradients were 98–80% A over 0–7 min, 80–78% A over 7–11 min, 78–40% A over 11–20 min, 40–35% A over 20–25 min, 25–28 min at 35% A, 35–5% A over 28–30 min, 30–33 min at 5% A, and final re-equilibration at 98% A for 5 min.

A Xevo G2-XS with an electrospray ionization source was used to collect MS data. MS was performed in both positive ion and negative ion modes under 30 V cone voltage, with a capillary voltage of 3.0 kV (positive ion mode) or 2.5 kV (negative ion mode). The desolvation temperature was set at 450 °C with a desolvation gas flow rate of

600 L/h, and the source temperature was set at 150 °C with a cone gas flow rate of 50 L/h. All data were collected in the $MS^E$ mode, with the following parameters: $MS^E$ range, 50–1200 $m/z$; $MS^E$ low energy, 6 eV; and $MS^E$ high energy, 15–30 eV. To calibrate the instrument, a sodium formate solution (0.5 mM) was used. Continuous acquisition of leucine enkephalin was used as an external standard for mass correction. All data were viewed in MassLynx v4.2 (Waters, Milford, MA, USA).

## Processing of metabolomics data
The raw data were first converted to the analysis base file (ABF Converter; https://www.reifycs.com/AbfConverter/) format before being imported into the MS-DIAL v4.60 software[93]. The parameter settings were as follows: the tolerances for MS1 and MS2 were 0.01 Da and 0.02 Da, respectively. The mass range of MS1 and MS/MS was set between 100 and 1000, with the MS/MS amplitude cutoff at 800. The retention time range was set between 1.0 and 29.5 min, with a retention time tolerance of 0.15 min. The width of the mass slice was 0.1 Da. Adduct types such as [M-H]⁻, [M + HCOO]⁻, [M+Na-2H]⁻, [M + K-2H]⁻, [2M-H]⁻, and [2 M + FA-H]⁻ were selected for the negative ion mode. [M + H]⁺, [M+Na]⁺, [M + K]⁺, [M + NH₄]⁺, and [2 M + H]⁺ were selected for the positive ion mode. Each of the obtained feature tables was split into two peak tables based on the retention time ranges, which were according to the accumulation level of phenolic acids (1.0–18.5 min for both root and leaf samples) and ATDs (16.5–29.5 min for root samples; 14.0–29.5 min for leaf samples). Normalized metabolomics data were imported into SIMCA-P 14.1 (Umetrics AB, Umea, Sweden) to conduct the chemometric analysis.

## Heterologous expression in yeast and yeast microsome isolation
Based on the genome and transcriptome sequencing data, all candidate CYP76AK genes were identified and cloned using the primers shown in Supplementary Data 12. The open reading frames were further subcloned into the epitope-tagged vector pESC-His using *Eco*R I and *Not* I restriction sites for expression in the WAT11 yeast strain, which contains *A. thaliana* NADPH-CYP reductase ATR1[94]. WAT11 transformed with empty pESC-His was employed as control. Transformants were cultured on SD dropout medium (-His) and grown at 28 °C for 72 h. A single positive colony was initially grown in 5 mL of SD-His liquid medium for about 24 h at 28 °C in a shaking incubator (200 rpm). The culture was then used to inoculate 250 mL of fresh SD-His liquid medium for another 24 h at 28 °C with shaking. Cells were then collected and washed three times with sterile water and yeast extract peptone dextrose medium with 2% galactose (YPL). Cells were then induced with YPL with shaking for 16 h at 28 °C. Tris-EDTA buffer solution was prepared with 50 mM Tris-HCl and 1 mM EDTA at pH 7.4. Cells were recovered and transferred to a 50-mL tube by centrifugation at 7000 × g for 5 min and were resuspended in 25 mL of TEK buffer (0.1 M KCl in TE buffer). Cells were then left at room temperature for 5 min and were again recovered and resuspended in pre-chilled TESB (0.6 M sorbitol in TE). All steps were then performed at 4 °C. Cells were lysed at 4–6 °C by a low-temperature ultra-high-pressure continuous-flow cell disrupter. The procedure was repeated three times. The tube was then centrifuged at 12,000 × g for 30 min, and the supernatant was transferred to tubes containing 10 mL polyethylene glycol (PEG)-NaCl (PEG4000 and 0.15 M NaCl). The tube was again centrifuged at 12,000 × g for 30 min. The supernatant was discarded, and the pellet was resuspended in TEG buffer (5% [v/v] glycerol in TE buffer).

## In vitro activity assays
Activity assays were performed in a 1.5-mL microtube using 500 μL of TE buffer (pH 7.5) that included 0.5 mg total microsomal proteins, 300 μM NADPH, 50 μM of substrate, and a regenerating system (2.5 μM FAD, 2.5 μM FMN, 1 mM DTT, 2 mM glucose-6-phosphate, and 2 U glucose-6-phosphate dehydrogenase). The reaction mixtures were incubated at 28 °C for 16 h in a shaking incubator (200 rpm) and then

were extracted with 500 μL of ethyl acetate three times with vortexing. After centrifugation at 12,000 × g for 10 min, the organic phase was transferred to fresh microtubes, concentrated under a vacuum, and resuspended in 200 μL methanol for LC-MS analysis.

## Reporting summary
Further information on research design is available in the Nature Portfolio Reporting Summary linked to this article.

## Data availability
The data supporting the findings of this work are available within the paper and the Supplementary Information files. A reporting summary for this article is available as a Supplementary Information file. The datasets generated and analyzed during this study are available from the corresponding author upon request. All the mass spectrometry data involved in this study have been deposited in National Omics Data Encyclopedia under Project ID: OEP004258. The transcriptome data involved in this study has been deposited on Genome Sequence Archive database (CRA010533) which is publicly accessible at https://ngdc.cncb.ac.cn/gsa. HMM files from Pfam were used to predict CYP450 (PF00067) gene families. Source data are provided with this paper.

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

## Acknowledgements

The authors thank Dr. David Nelson (The University of Tennessee Health Science Center) of P450 nomenclature committee for the naming of the P450s. We thank Mr. Jun Zhang from Kunming Plant Classification Biotechnology Co., Ltd. for the collection and identification of species. We also thank Dr. Pan Liao from for Hong Kong Baptist University for valuable advice on the manuscript. This work was financially supported by the National Key R&D Program of China (2022YFC3501700), National Natural Science Foundation of China (31970325, 32070327), Young Elite Scientists Sponsorship Program by Cast (2021-QNRC1-02), and Research Project of Science and Technology Commission of Shanghai Municipality (21DZ2202300).

## Author contributions

W.S.C., J.F.C., S.Q., and J.D.H. planned and designed the research. J.D.H., F.Y.W., R.J., J.X.L., Z.Z., S.Y.F., Y.Q.Q., J.D., Y.J. prepared the plant material for sequencing and chemistry profiling. C.-L.X. identified all species. Z.C.X., Q.L., and P.D. performed the bioinformatics analyses. S.Q. and F.Y.W. analyzed the metabolic data. J.D.H. Z.D.W., J.W., X.X.W., and Y.C.Z. carried out experiments. J.D.H., F.Y.W., Z.C.X., Q.L., S.Q., J.F.C., and W.S.C. interpreted the data and participated in the discussion. J.D.H., Z.C.X., S.Q., and J.F.C., wrote the paper. W.S.C. revised the paper. The authors J.D.H., S.Q., F.Y.W., and Q.L. contributed equally to this work.

## Competing interests

The authors declare no competing interests.
