## [Peer Review File · Nature Communications]

Functional divergence of CYP76AKs shapes the chemodiversity of abietane-type diterpenoids in genus *Salvia*Reviewer #1 (Remarks to the Author):

In their article "Lineage-wide metabolic and enzymic profiling reveal the role of CYP76AKs in chemodiversity of abietane-type diterpenoids in *Salvia* (Lamiaceae, Nepetoideae)", Hu et al. combine phylogentic and phylogenomic methods with mass-spec-based profiling of specialized metabolites in *Salvia* in order to trace the evolution of their chemodiversity. Overall, the paper is of extremely high quality and the results are novel. This will be one of the prime examples that I will use in the future for plants' evolution of chemodiversity.

I just have a few remarks for the authors to consider.

Major comments:

(A) "secondary metabolites"  this term has gone a bit out of fashion in the community. Please use specialized metabolite throughout.

(B) Line 63 to 74: Here a bit more of a broad picture perspective is necessary. Indeed, such analyses are now possible and have been extensively used to reconstruct deep evolutionary events in the origins of plant specialized metabolism, see and cite:

<https://doi.org/10.1016/j.semcdb.2022.03.004>

(C) Along these lines a text on the evolution and diversity of CYP450 enzymes in plants would be prudent, e.g., D. Nelson, D. Werck-Reichhart, A P450-centric view of plant evolution: P450-centric evolution, *The Plant Journal*. 66 (2011) 194–211. <https://doi.org/10.1111/j.1365-3113.2011.04529.x>.

(D) Please upload all alignments as supplementary files

(E) Please deposit all mass spectrometry data on <https://www.ebi.ac.uk/metabolights/>

(F) The colors in Figure 3 are confusing, because some of the colors for the species clades (i.e. the clades based on Figure 1) are very similar to the colors of the CYP gene family clades

(G) The RNAseq data needs a bit more explanation. What is exactly plotted in Figure 5? What is this relative to? The average expression in roots and leaves? The material and methods section on this also does not answer my questions and needs more details on how these values were calculated.

Minor comments:

There are a couple of grammar mistakes.

Reviewer #2 (Remarks to the Author):

The submitted article entitled "Lineage-wide metabolic and enzymic profiling reveal the role of CYP76AKs in chemodiversity of abietane-type diterpenoids in *Salvia* (Lamiaceae, Nepetoideae)" by Hu, Wang, Zu, Li, et al is an impressive, comprehensive and robust examination of a subfamily of enzymes with a role in specialised metabolite pathway.

The authors ultimately set out to see how and why abietane-type diterpenoids (ATDs) varied across the *Salvia* genus. They sequenced 71 new transcriptomes (!) and performed LCMS metabolite analysis to see what ATDs were present in what species/organ. They found clear distribution of subtypes across phylogenetic clades and organs. Then they assessed the CYP76AK subfamily previously known to be involved in the C20 modification that was important for the ATD pathway branching. Multiple enzymes were cloned and tested in yeast, and new activities identified. Crucially the activity and gene distribution matched the metabolomics work (with the C20 modification being important). Then to get an evolutionary picture they performed ancestral sequence/state reconstruction to reveal that the ancestors could probably make all ATDs and their CYP76AK perform the full carboxylation reactions. Only through "loss of function" of the CYP76AKs

did the ATD diversity originate, i.e. by the enzymes stopping at C20 hydroxylation which opened the door to tanshinone biosynthesis. This "loss of function" is, apparently, an unique form of metabolic evolution.

The authors use a wide array of method to investigate their research question including transcriptome sequencing, metabolomics, comparative genomics, enzyme assays, and ancestral state/sequence reconstruction. The data and methods are robust and very well presented and documented.

The approach is fairly novel though clearly very inspired in method and appearance by work conducted on the whole mint family by the mint genomics consortium and on *Nepeta* by Lichman et al. However taking this approach comprehensively into a whole genus and focussing on an enzyme subfamily is certainly new and worthy of wide attention. Furthermore, the results are very interesting, and they reach conclusions using multiple complementary methods.

Whilst I fully endorse the quality of the work, its potential influence for other studies and its multi-disciplinary approach, I have some general criticisms that need to be addressed (see below). I have also added comments to the PDF document regarding wording and sentence structures.

General points

- The known pathway needs to be clearer. Preferable add a figure showing the *Salvia* ATDs pathway including enzymes involved.
- The lack of knowledge about how the 20-hydroxylation leads to demethylation must be clearer in the introduction, and then mentioned again in the discussion. This paper implicates CYP76AK evolution as very important for ATD diversity but of course, there might be (yet unknown) enzymes evolving to do the demethylation that could be as important.
- There are multiple causal statements – e.g. this X is a driving force for Y, or this function change caused the chemical distribution. These are generally just correlations and the subjective causality aspect must be reserved for the discussion. I have marked these in the PDF.
- I disagree with the notion of "loss of function" here, and the nature of the 3 step oxidation is not properly discussed. The *in vitro* enzyme work really refers to loss of activity rather than loss of function. To establish function the enzymes really need to be validated *in vivo*, which is beyond the scope of this work. Please modify the language accordingly. Also, is it really a loss? This is subjective. We start with a promiscuous enzyme that makes 3 different products and fails to properly control the 3-step oxidation. In contrast, the CYP76AK1 makes a single product to a high level with few side products. Maybe CYP76AK1 has gained specificity and catalytic efficiency. The whole focus of the work on "loss of function" is simplistic and the subtle complexity at play here must be discussed further.
- The P450 3 step methyl oxidation to carboxylic acids is known in other systems (e.g. artemisinin, iridoids) but there are helper enzymes that contribute to the latter 2 oxidation steps. P450s are not very efficient at all 3 steps, and here we see accumulating intermediates in the 3 step reactions (not just the carboxylate). Can this be discussed? It relates to the "loss of function" point above, as it hints at an inefficient enzyme.
- The ancestors are only tested with hydroxysugiol but the tanshinone precursor is ferruginol. Is this OK – do you think the activities on the ferruginol substrates will be the same? Why were the ancestors not tested together with AH3 to look at ferruginol and the tanshinone pathway? This needs explanation or justification.
- There are notable side products with AK1 and AK22 at EIC 329 with hydroxysugiol – do you know what these are? The possibility of CYPs making other compounds is important to understand their evolution.
- Please add compound names to Figure 2
- Fig 6 D and E needs clearer information about what the dots/colours mean.
- Can the authors clarify the species tree approach – are "five sets of OGs, composing of 2178, 1532, 1169, 512, and 130 orthologous genes" 5 different orthogroups or 5 sets of OGs. If the latter how did you choose those numbers and which OGs were included and why.

See further minor comments in the PDF.

In their review of the first version of this manuscript, reviewer #2 added some comments to the manuscript file. These comments were forwarded to the authors, who replied as included in this Peer Review File.

Thank you for arranging reviews for our manuscript (number: NCOMMS-23-02768). We are pleased to know that our study is of general interest for the readers of *Nature Communications*.

We have carefully evaluated the critical comments and thoughtful suggestions made by the editor and reviewers, responded to them point-by-point, and revised the manuscript accordingly (marked in red to show changes). Moreover, we tried our best to improve the manuscript and made some changes in the manuscript. These changes will not influence the content and framework of the paper. And here we did not list the changes but marked in red in revised paper.

Our responses to each of the comments and suggestions are listed below.

Reviewer #1:

Major comments:

Comment 1: "secondary metabolites"  this term has gone a bit out of fashion in the community. Please use specialized metabolite throughout.

The authors' response: We think this is an excellent suggestion. We agree that using "specialized metabolites" would be more appropriate than "secondary metabolites" throughout the manuscript. We have made the changes at **lines 50, 53 and 78**.

Comment 2: Line 63 to 74: Here a bit more of a broad picture perspective is necessary. Indeed, such analyses are now possible and have been extensively used to reconstruct deep evolutionary events in the origins of plant specialized metabolism, see and cite: <https://doi.org/10.1016/j.semcdb.2022.03.004>

The authors' response: We quite agree with this comment. The reference recommended by the reviewer comprehensively showcased various aspects of

metabolic pathways that evolve along with the speciation of plant kingdom, which would help to enhance the meaning of this study. In the revised manuscript (**lines 62-77**), we have expanded the significances and purposes of a lineage scale studies on plant metabolites and their biosynthetic pathways. We also gave more in-depth description on the instances we cited.

Comment 3: Along these lines a text on the evolution and diverstiy of CYP450 enzymes in plants would be prudent, e.g., D. Nelson, D. Werck-Reichhart, A P450-centric view of plant evolution: P450-centric evolution, *The Plant Journal*. 66 (2011) 194–211. <https://doi.org/10.1111/j.1365-313X.2011.04529.x>.

The authors' response: Thanks a lot for the suggestion, which reminds that we need to clarify the significance of this study for the understanding of CYP evolution. However, in order to keep the rhythm of the text, we are not planning to immediately describe the significance of CYP evolution just after the section on the meaning of plant metabolism on lineage scale. Instead, we included a discussion on the significance of our research in understanding the evolution of plant CYPs in the section introducing the biosynthetic pathway of tanshinones (**lines 83-106**).

Comment 4: Please upload all alignments as supplementary files

The authors' response: Thank you so much for pointing this out. All the alignments used for the phylogenetic analyses are provided as the source data in the supplementary material.

Comment 5: Please deposit all mass spectrometry data on <https://www.ebi.ac.uk/metabolights/>

The authors' response: We sincerely appreciate the valuable comments. All mass spectrometry data, including metabolomics and enzymatic activity evaluation data,

have been uploaded and deposited at <https://www.ebi.ac.uk/metabolights/MTBLS7637/files>, which will release around 2023-05-25. Moreover, the transcriptome data involved in this study were deposited on Genome Sequence Archive database (<https://ngdc.cncb.ac.cn/gsa/>) under accession nos. CRA010533 (<https://ngdc.cncb.ac.cn/gsa/s/5O91gbq9>).

MTBLS7637: Lineage-wide metabolic and enzymatic profiling reveal the role of CYP76AKs in chemodiversity of abietane-type diterpenoids in *Salvia* (Lamiaceae, Nepetoideae)

Status **In Curation** Release Date **2023-05-25**

SUMMARY

The genus *Salvia* L. (Lamiaceae) comprises myriad distinct medicinal herbs, with terpenoids as one of their major active chemical groups. Abietane-type diterpenoids (ATDs), such as tanshinones and carnosic acids, are specific to *Salvia*, with characteristic chemical diversity among lineages. To elucidate how ATDs chemical diversity evolved, we undertook large-scale metabolic and phylogenetic analyses of 71 *Salvia* species, combined with enzyme function, ancestral sequence and chemical trait reconstruction, and comparative genomics experiments. This integrated approach showed that the lineage-wide ATDs variations in *Salvia* were induced by differences in the oxidation of the terpenoid skeleton at C-20, which was caused by the loss of a function of the cytochrome P450 subfamily CYP76AK. These findings present a unique pattern of chemical diversity in plants that was driven by the loss of enzyme activity and associated catalytic pathways.

 Guided submission

 Study overview

Comment 6: The colors in Figure 3 are confusing, because some of the colors for the species clades (i.e. the clades based on Figure 1) are very similar to the colors of the CYP gene family clades

The authors' response: We appreciate your valuable comments. We acknowledge that the colors used in **Fig. 3** for gene family clades are similar to those used for the species clades, which can cause confusion. To address this issue, we have modified the figure by changing the colors of the gene family clades to alternating black and gray. This will clearly distinguish the gene family clades from the species clades and improve the readability of **Fig. 3** in the manuscript.

Fig. 3. Cladogram of the CYP76AK subfamily members. The branch color represents the lineage which the species containing these CYP76AK genes belong. See **Supplementary Fig. 32** for the annotated phylogram. PR, subgenera *Perovskia* and *Rosmarinus*. Source data are provided as a Source Data file.

Comment 7: The RNAseq data needs a bit more explanation. What is exactly plotted in Figure 5? What is this relative to? The average expression in roots and leaves? The material and methods section on this also does not answer my questions and needs more details on how these values were calculated.

The authors' response: Thank you so much indeed for your suggestions. We have incorporated your suggestion by making revisions to the Material and Methods section (**lines 553 to 560 and lines 569 to 574**) and the legend of **Fig. 5**. Furthermore, we have reorganized **Fig. 5** to enhance the clarity of the information presented.

Fig. 5. Expression profiles of the CYP76AK subfamily members in *Salvia*. Gene expression heatmap for CYP76AK subfamily members from representative species of each branch of *Salvia*, displaying the relative expression levels in leaves (L) and roots (R). The heatmap shows three biological replicates, each plotted individually. Expression values represent Z-score-transformed log (TPM + 1) values. Species abbreviations can be found in **Supplementary Table 1**. Source data are provided as a Source Data file.

Minor comments:

Comment 8: There are a couple of grammar mistakes.

The authors' response: We appreciate the reviewer's feedback and have carefully reviewed the manuscript annotations to make necessary modifications (**marked in**

red in the revised manuscript).

Reviewer #2:

General points

Comment 1: The known pathway needs to be clearer. Preferable add a figure showing the *Salvia* ATDs pathway including enzymes involved.

The authors' response: We sincerely appreciate the valuable comments. We acknowledge the significance of the reported biosynthetic pathways of ATDs in *Salvia* and the importance of visualizing them in a clear and concise manner. In response, we have added a new figure, **Supplementary Fig. 1** in the supplementary material, which summarizes the *Salvia* ATDs biosynthetic pathways. We have also included the corresponding functional enzymes and key steps, marked for clarity.

Supplementary Fig. 1. Proposed ATDs biosynthetic pathway in the genus *Salvia*²⁵⁻³⁰. Based on the biosynthesis study of ATDs in *Salvia* species (*S. pomifera*, *S. fruticose*, *S. rosmarinus*, *S. officinalis*, and *S. miltiorrhiza*), miltiradiene is derived from GGPP and catalyzed by CPS and KSL terpene synthases, which spontaneously oxidizes to abietatriene. The CYP76AH1 enzyme (green) then oxidize abietatriene to the precursor ferruginol, while the 11-hydroxyferruginol synthase (CYP76AH3-4/22-24, green) further oxidizes ferruginol to 11-hydroxyferruginol, 7,

11-dihydroxyferruginol, sugiol and 11-hydroxysugiol. In addition, CYP76AK1 (grey) catalyzes oxidation at C-20 to generate an alcohol group, while CYP76AK6-8 (red) carry out hydroxylation, carbonylation, and carboxylation at C-20 to oxidize ferruginol and 11-hydroxyferruginol to pisiferic acid and carnosic acid, respectively. However, the mechanism of coupled demethylation (C20) and aromatization (ring B) reactions to form miltirone, as well as the demethylation (C19) and aromatization of ring A, remain unclear. CYP71D375 (orange) is known to generate the 14, 16-epoxy D-ring, which is then further oxidized to a furan ring by 2ODD14 (yellow) during tanshinones biosynthesis. The citation of the literatures is consistent with the manuscript.

Comment 2: The lack of knowledge about how the 20-hydroxylation leads to demethylation must be clearer in the introduction, and then mentioned again in the discussion. This paper implicates CYP76AK evolution as very important for ATD diversity but of course, there might be (yet unknown) enzymes evolving to do the demethylation that could be as important.

The authors' response: We appreciate your insightful comments. The demethylation and aromatization reactions have been the focus of interest in the biosynthesis of terpenoids, particularly in the formation of tanshinones. Previous studies suggested that carnosic acid, with C20 oxidized to a carboxylic acid, could be the potential substrate for the formation of miltirone, triggering aromatization via the formation of a heterocyclic bridge between C20 and C7. However, based on our metabolomics analysis of *Salvia*, we found that the accumulation pattern of carnosic acid and the expression pattern of corresponding enzymes CYP76AK6/7/8 were inconsistent with tanshinones, indicating that carboxylation on C20 is not the upstream reaction of demethylation and aromatization. Instead, hydroxylation (C20) catalyzed by CYP76AK1/22 showed a stronger correlation with miltirone, the key precursor for the formation of diverse tanshinones, suggesting that it may act as a key upstream reaction for the coupled demethylation and aromatization. Therefore, we have revised the Introduction (**lines 98-100**) and Discussion (**lines 476-487**) sections to include this

finding. We appreciate your feedback and hope that these modifications have addressed your concerns.

Comment 3: There are multiple causal statements – e.g. this X is a driving force for Y, or this function change caused the chemical distribution. These are generally just correlations and the subjective causality aspect must be reserved for the discussion. I have marked these in the PDF.

The authors' response: Thank you for bringing this issue to our attention. We greatly appreciate your valuable feedback. We agree that the term "driving force" implies a strong causality, and we acknowledge that correlation cannot be equated with causation. Therefore, we have carefully revised all instances of the phrase "driving force" in our manuscript to more accurately reflect the observed correlations. Instead, we have used phrases such as "correlated with" and "key factor" (**lines 428 and 512**) to describe the relationships between the variables.

Comment 4: I disagree with the notion of “loss of function” here, and the nature of the 3 step oxidation is not properly discussed. The in vitro enzyme work really refers to loss of activity rather than loss of function. To establish function the enzymes really need to be validated in vivo, which is beyond the scope of this work. Please modify the language accordingly. Also, is it really a loss? This is subjective. We start with a promiscuous enzyme that makes 3 different products and fails to properly control the 3-step oxidation. In contrast, the CYP76AK1 makes a single product to a high level with few side products. Maybe CYP76AK1 has gained specificity and catalytic efficiency. The whole focus of the work on “loss of function” is simplistic and the subtle complexity at play here must be discussed further.

The authors' response: We agree with this comment. “Loss of activity” is indeed a more accurate statement. We have replaced “loss of function” by “loss of activity” across the whole manuscript. However, we insist on the statement that CYP76AK1

does not execute carbonylation and carboxylation activities on C-20 rather than reduced catalytic activities. Firstly, CYP76AK1 and 22 demonstrated no detectable carbonylation and carboxylation activities. Secondly, although the in vivo functional studies were not performed, this statement can be supported by metabolic profiles from another perspective, for we can't detect any ATDs with carbonylation or carboxylation at C-20 from species in Clade IV. And this is a lineage wide pattern, not an individual phenomenon. Taken together with the in vitro enzyme experiments, we believe it is sufficient to support the judgment of loss of activity. Additionally, our result only focused the divergence in oxidative activities towards C-20. In fact, we also discovered CYP76AK1 and 22 have the capacity of hydroxylation on C-6 (**Supplementary Fig. 38**), which indicated a more complex functional divergence of CYP76AKs, and arose further topic of research.

Comment 5: The P450 3 step methyl oxidation to carboxylic acids is known in other systems (e.g. artemisinin, iridoids) but there are helper enzymes that contribute to the latter 2 oxidation steps. P450s are not very efficient at all 3 steps, and here we see accumulating intermediates in the 3 step reactions (not just the carboxylate). Can this be discussed? It relates to the “loss of function” point above, as it hints at an inefficient enzyme.

The authors' response: Thank you for the suggestion that should not be ignored. To address this question, we consulted literatures to see the catalytic properties of similar CYPs that can catalyze continuous oxidation reactions. Take the biosynthesis of artemisinin as instance, using an in vitro microsomes system, CYP71AV1 catalyzes the oxidation of amorpha-4,-11-diene at C-12 to form artemisinic acid, via alcohol and aldehyde intermediates (Teoh et al., 2006). However, alcohol dehydrogenase (ADH1) and artemisinic aldehyde dehydrogenase (ALDH1) were proved to contribute in artemisinin biosynthesis by catalyzing the artemisinic aldehyde and artemisinic acid formation of in trichomes of *A. annua* (Teoh et al., 2009; Paddon et al., 2013). This example illustrates that the in vitro enzymatic evidences may not reflect the real

metabolic process accurately in plant cells. Similarly, in addition to CYP76AKs, there may also be other enzymes involved in catalyzing the oxidation of ATDs at C-20. However, this needs further in vivo activity verifications of CYP76AKs and exploration of enzymes as ALDH1 in *A. annua*. Based on your suggestions, we have added a depth discussion at **lines 491-501**.

Comment 6: The ancestors are only tested with hydroxysugiol but the tanshinone precursor is ferruginol. Is this OK – do you think the activities on the ferruginol substrates will be the same? Why were the ancestors not tested together with AH3 to look at ferruginol and the tanshinone pathway? This needs explanation or justification.

The authors' response: Thank you for bringing this issue to our attention. As depicted in **Fig. 4**, we utilized ferruginol and 11-hydroxysugiol as substrates to evaluate the enzyme activity of CYP76AKs. Our results showed that the products produced by the tested CYP76AKs were identical for both substrates. In response to your comment, we have retested the enzyme activity of all six ancestral genes with ferruginol in combination with CYP76AH3. The revised figure (**Fig. 6**) includes the additional data, which confirms that the catalytic functions of the six ancestral genes on both substrates are consistent (**Fig. 6c and 6d**). We appreciate your suggestion and have incorporated the necessary changes to improve the clarity of our results.

Fig. 6c and 6d. Ancestral function of CYP76AKs. Extracted ion chromatograms (EICs) showing *in vitro* catalytic activity of each ancestral CYP76AK using ferruginol in combination with CYP76AH3 and 11-hydroxysugiol as the substrates.

Comment 7: There are notable side products with AK1 and AK22 at EIC 329 with hydroxysugiol – do you know what these are? The possibility of CYPs making other compounds is important to understand their evolution.

The authors' response: Thank you for bringing this to our attention. As depicted in Figure 4, SmCYP76AK1 and Sc76AK22 catalyzed a product appearing at 17.25 min with an extracted ion chromatogram (EIC) of 329.175. We investigated two additional CYP76AK22 genes derived from *S. apiana* and *S. pachyphylla* but they did not catalyze this product, indicating that it may be a specific side product of SmCYP76AK1 and ScCYP76AK22 (**Supplementary Fig. 38a**). To determine the structure of this side product, we analyzed the characteristic neutral loss and compared the MS behavior using analogues. The 18 Da neutral loss instead of 30 Da suggests that the hydroxyl group is on the ring rather than the methyl group, which distinguishes it from the main product (EIC: 331.190; 11, 20, -dihydroxysugiol). In addition, the 2 Da difference between the main and side products suggests that the

ortho-hydroxyl group on C-11 and C-12 may have been spontaneously oxidized as C-11, C-12 keto abietane (**Supplementary Fig. 38b**). Based on this structural feature, we compared the fragmentation patterns of the isomeric analogue (EIC: 329.175) with this side product, which revealed that both compounds have the same MS behavior as continuous neutral loss of two methyl groups. The double bonds at C-5 and C-6 of the isomeric analogue may make it difficult for the hydroxyl group at C-6 to undergo neutral loss, while the lower intensity of the ion (311.168) by losing hydroxyl group indicates that the hydroxyl group may locate at the C-6 position for the side product due to the influence of carbonyl group at the C-7 position. The tentatively identified structure of the side product is shown in **Supplementary Fig. 38b**. These findings suggest that some CYP76AKs may have other side functions, offering more possibilities for the chemical diversity of ATDs in *Salvia* and warranting further functional exploration

Supplementary Fig. 38. Structure elucidation of side product catalyzed by SmCYP76AK1 and ScCYP76AK22. (a) Extracted ion chromatogram (EIC) analysis of the CYP76AK1/22 catalytic side product *in vitro* with 11-hydroxyferruginol along with an isomeric analogue; (b) Structure elucidation of the side product based on the characteristic neutral loss and compared to the isomeric analogue.

Comment 8: Please add compound names to Figure 2.

The authors' response: Thank you for your valuable suggestion. We have addressed it by adding the compound names to the legend of **Fig. 2** instead of directly on the figure itself. This approach allows us to provide the necessary information without cluttering the figure. We have also ensured that the compound names in the legend are consistent with the compound names listed in **Supplementary Table 8** in the supplementary material. The revised legend of **Fig. 2** was shown below:

Fig. 2. Chemical diversity and distribution of ATDs in *Salvia*. The most prevalent ATDs identified by LC-MS in both roots and leaves from *Salvia*. (a) Thirty-six ATDs were divided into five groups based on different structural characteristics. ATDs in groups A, C, D, and E (which accumulated specifically in leaves) exhibited higher intensities in the negative ion mode, and ATDs in group B (which accumulated specifically in roots) exhibited higher intensities in the positive ion mode. **1:** 7, 20-Dihydroxyabietaquinone; **2:** 16, 20-Dihydroxyabietaquinone; **3:** 7-Keto, 20-hydroxyabietaquinone; **4:** 11, 20-Dihydroxysugiol; **5:** 20-Hydroxyabietaquinone; **6:** Miltirone; **7:** Cryptotanshinone; **8:** Tanshinone IIA; **9:** 1, 2-Dihydrotanshinquinone; **10:** Dihydrotanshinone I; **11:** Methylene-tanshinquinone; **12:** Tanshinone I; **13:** Przewaquinone C; **14:** 7-Keto-royleanone; **15:** Royleanone; **16:** 11-Hydroxysugiol; **17:** 7-Hydroxyroyleanone; **18:** 6-Hydroxyroyleanone; **19:** 6-Keto-11-hydroxysugiol; **20:** 7, 11-Dihydroxyferruginol; **21:** 11, 14-Dihydroxysugiol; **22:** 11-Hydroxyferruginol; **23:** 20-Keto-2, 11-dihydroxyferruginol; **24:** 20-Keto-11-hydroxysugiol; **25:** 20-Keto-7-hydroxyferruginol; **26:** 20-Keto-6-hydroxyferruginol; **27:** 20-Keto-7-acetyferruginol; **28:** 7-Hydroxy-20-carboxyabietaquinone; **29:** 7-Hydroxyisocarnosol; **30:** 6-Hydroxycarnosol; **31:** 7-Keto-carnosic acid; **32:** 7-Methoxycarnosol; **33:** 6-Methoxycarnosol; **34:** Carnosol; **35:** Carnosic acid; **36:** Methyl carnosate. (b) Distribution patterns of the 36 ATDs in 71 *Salvia* species. PR, subgenera *Perovskia* and *Rosmarinus*. Source data are provided as a Source Data file.

Comment 9: Fig 6 D and E needs clearer information about what the dots/colours

mean.

The authors' response: Thank you for bringing this to our attention. In **Fig. 6**, the dots represent the predicted ancestral metabolic traits, while the colors indicate the ability to produce carnosic acid (green), tanshinones (red), and 20-keto ATDs (light blue). As for **Fig. 6e and 6f**, it can be concluded that the ancestors of *Salvia* likely produced all three key traits. However, with the speciation of *Salvia*, the retention of all metabolic traits was observed in *S. rosmarinus* and partial Clade II (North American taxa), while partial loss was observed in other lineages (Clade I, Clade IV, and partial Clade II). Additionally, based on the results of enzyme activity assays of CYP76AKs derived from ancestors and different clades, the loss of activity of CYP76AKs suggests a correlation with the metabolic diversity among lineages. Based on this, we have rearranged and revised the **Fig. 6** to make the information clearer.

Fig. 6. Evolution of the CYP76AK subfamily. (a) Syntenic relationships among *CYP76AK* genes in *Salvia* genomes revealed the divergence of *CYP76AK* clades. The phylogenetic tree in the left panel represents the phylogeny of six *Salvia* species with *N. cataria* as the outgroup. WGD events are indicated by red dots. Microsynteny of *CYP76AK* genes in the synteny blocks. Red polygons show syntenic relationships between *CYP76AK* genes, and gray polygons show other

genes. Anc-CYP76AK, Ancestral *CYP76AK* gene. (b) Reconstruction of ancestral CYP76AK proteins. Ancestral nodes (ancNodes) 1–6 show the resurrected enzyme of each major branch point of the CYP76AK phylogenetic tree. Functional characters of each branch are distinguished by different colors. (c) and (d) Ancestral function of CYP76AKs. Extracted ion chromatograms (EICs) showing *in vitro* catalytic activity of each ancestral CYP76AK using 11-hydroxyferruginol and 11-hydroxysugiol as the substrate, respectively. (e) Evolution of ATDs biosynthesis in *Salvia*. The phylogenetic tree shows the speciation of the clades of *Salvia*, and the diamonds represent the predicted ancestral metabolic traits for each speciation node. Square frames represent the realistic metabolic traits of each *Salvia* clade as shown in Fig. 3. Green, red, and light blue represent the existence of carnosic acid, tanshinones, and 20-keto ATDs, respectively. The time scale on the left reflects the time of speciation for each clade. Branches highlighted in light blue indicate clades in which the ancestral metabolic traits have been retained. (f) Chronology of *CYP76AK* genes. The phylogenetic tree represents the evolution of CYP76AK clades according to the MCMCtree model. The speciation time for each clade is labeled. Branches highlighted in blue indicate CYP76AK clades that have retained their ancestral catalytic functions, and branches highlighted in red indicate clades that have undergone a loss of function. The arrows at the bottom indicate the CYP76AK clade that contributes to producing the type of ATDs in the corresponding lineage. Green, red, and indigo gray arrows indicate the biosynthesis of carnosic acid, tanshinones, and 20-keto ATDs, respectively. Source data are provided as a Source Data file.

Comment 10: Can the authors clarify the species tree approach – are “five sets of OGs, composing of 2178, 1532, 1169, 512, and 130 orthologous genes” 5 different orthogroups or 5 sets of OGs. If the latter how did you choose those numbers and which OGs were included and why.

The authors’ response: Thank you for your helpful comments. In this study, we selected five different gene sets from the orthogroup of all 77 species to construct phylogenetic trees. We focused on utilizing highly conserved single-copy genes or low-copy genes that exhibit bi-parental inheritance, which are known to be effective in addressing the hybridization, speciation, and incomplete lineage sorting of closely

related species (10.1111/j.1469-8137.2012.04212.x). Specifically, the 130 orthogroups represented the number of single-copy genes present in all tested species. The remaining gene sets were selected based on the gene copy numbers and taxon coverage, which included 512 orthogroups representing all single-copy genes and low-copy genes with less than two copies covering 50% of taxa, 1169 orthogroups representing all single-copy genes and low-copy genes with less than two copies in all tested species, 1532 orthogroups representing all single-copy genes and low-copy genes with less than three copies covering at least 50% of taxa, and 2178 orthogroups representing all single-copy genes and low-copy genes with less than three copies in all tested species. We believe that these gene sets will provide valuable insights into the phylogenetic relationships of the species under study.

Reviewer #1 (Remarks to the Author):

In this revised version of the manuscript (which was already of very high quality upon initial submission), the authors have satisfactorily addressed all my remarks. I do not have any further comments and congratulate the authors on a very nice manuscript.

Reviewer #2 (Remarks to the Author):

All my comments were dealt with. Congratulations to the authors for their excellent work.

REVIEWERS' COMMENTS

Reviewer #1 (Remarks to the Author):

In this revised version of the manuscript (which was already of very high quality upon initial submission), the authors have satisfactorily addressed all my remarks. I do not have any further comments and congratulate the authors on a very nice manuscript.

The authors' response: We sincerely appreciate your positive feedback and recognition of our efforts in addressing your previous comments. We are delighted to hear that the revised manuscript meets your expectations and we thank you for your kind words.

Reviewer #2 (Remarks to the Author):

All my comments were dealt with. Congratulations to the authors for their excellent work.

The authors' response: We are grateful for your feedback and for acknowledging that we have effectively addressed all of your comments. Thank you for recognizing our work as excellent. We appreciate your support and encouragement.